# Toward resolving the budget discrepancy of ozone-depleting carbon tetrachloride (CCl$_4$): An analysis of top-down emissions from China

Sunyoung Park[1,2], Shanlan Li[2,3], Jens Mühle[4], Simon O'Doherty[5], Ray F. Weiss[4], Xuekun Fang[6], Stefan Reimann[7], Ronald G. Prinn[6]

[1]Department of Oceanography Kyungpook National University, Daegu 41566, Republic of Korea
[2]Kyungpook Institute of Oceanography Kyungpook National University, Daegu 41566, Republic of Korea
[3]Climate Research Division, National Institute of Meteorological Sciences, Seogwipo, Korea
[4]Scripps Institution of Oceanography, University of California, San Diego, La Jolla, CA 92093, USA
[5]School of Chemistry, University of Bristol, Bristol, UK
[6]Center for Global Change Science, Massachusetts Institute of Technology, Cambridge, MA 02139, USA
[7]Empa, Laboratory for Air Pollution and Environmental Technology, Swiss Federal Laboratories for Materials Science and Technology, Überlandstrasse 129, 8600 Dübendorf, Switzerland

*Correspondence to*: Sunyoung Park (sparky@knu.ac.kr)

**Abstract.** Carbon tetrachloride (CCl$_4$) is a first-generation ozone-depleting substance, and its emissive use and production were globally banned by the Montreal Protocol with a 2010 phase-out; however, production and consumption for non-dispersive use as a chemical feedstock and as a process agent are still allowed. This study uses the high frequency and magnitude of CCl$_4$ pollution events from an 8-year real-time atmospheric measurement record obtained at Gosan station (a regional background monitoring site in East Asia) to present evidence of significant unreported emissions of CCl$_4$. Top-down emissions of CCl$_4$ amounting to $23.6 \pm 7.1$ Gg yr$^{-1}$ from 2011 to 2015 are estimated for China, in contrast to the most recently reported, post-2010, Chinese bottom-up emissions of 4.3–5.2 Gg yr$^{-1}$. The missing emissions (~19 Gg yr$^{-1}$) for China contribute to approximately 54% of global CCl$_4$ emissions. It is also shown that $89 \pm 6\%$ of CCl$_4$ enhancements observed at Gosan are related to CCl$_4$ emissions from the production of CH$_3$Cl, CH$_2$Cl$_2$, CHCl$_3$, and C$_2$Cl$_4$ (PCE) and its usage as a feedstock and process agent in chemical manufacturing industries. Specific sources and processes are identified using statistical methods, and it is considered highly unlikely that CCl$_4$ is emitted by dispersive uses such as old landfills, contaminated soils, and solvent usage. It is thus crucial to implement technical improvements and better regulation strategies to reduce evaporative losses of CCl$_4$ occurring at the factory and/or process level.

## 1. Introduction

Carbon tetrachloride (CCl$_4$) is a long-lived greenhouse gas and an ozone-depleting substance. Its emissive use, production, and consumption are regulated under the Montreal Protocol on Substances that Deplete the Ozone Layer and its Amendments (MP). After reaching a peak in the early 1990s, the atmospheric abundance of CCl$_4$ has been decreasing at a rate of $-4.9 \pm 0.7$ ppt Cl yr$^{-1}$ (Carpenter et al., 2014) due to the phase-out of CCl$_4$ use in MP non-Article-5 (developed) countries by 1995. MP Article-5 (developing) countries, including China, were required to cease CCl$_4$ production and consumption for dispersive applications by 2010. However, CCl$_4$ production and consumption for non-dispersive use (e.g., as chemical feedstock and as a process agent) continues to be allowed, and thus CCl$_4$ is still produced and consumed alongside the increasing production of non-ODS chemicals (Carpenter et al., 2014). At present, the global bottom-up CCl$_4$ emissions derived from reporting countries are 3 (0–8) Gg yr$^{-1}$ for 2007–2013 (Carpenter et al., 2014; Liang et al., 2016).

The recent SPARC report (Liang et al., 2016) updated bottom-up anthropogenic CCl$_4$ emissions to at most 25 Gg yr$^{-1}$ in 2014, based on re-consideration of industrial production processes plus usage (15 Gg yr$^{-1}$), and the upper-limit estimate of 10 Gg yr$^{-1}$ for the potential escape from legacy sites and unreported inadvertent emissions (Sherry et al., 2017).

To verify these bottom-up estimates, independent top-down $CCl_4$ emission studies have used the total lifetime of $CCl_4$ with atmospheric observations (i.e., the observed decline rate of $CCl_4$ concentrations) and atmospheric transport models to derive "top-down" emission estimates. Using the most current estimates for the lifetime of $CCl_4$ in the atmosphere, soil, and ocean (Liang et al., 2016; Rhew and Happell, 2016; Butler et al., 2016), global top-down emissions to the atmosphere were calculated as $40 \pm 15$

Gg yr$^{-1}$ from 2007 to 2014 (Liang et al., 2016). A recent top-down study based upon the observed temporal trend and inter-hemispheric gradient of atmospheric $CCl_4$ (Liang et al., 2014) consistently derived global $CCl_4$ emissions of $30 \pm 5$ Gg yr$^{-1}$ from 2000 to 2012 when using the newly determined relative strength of oceanic sink versus soil loss (Liang et al., 2016). Therefore, the best estimate of global emissions from top-down methods is $35 \pm 16$ Gg yr$^{-1}$, which is significantly higher than reported emissions of 3 Gg yr$^{-1}$, even when considering large uncertainties relating to soil and ocean $CCl_4$ sinks (and how those sinks might

change over time). Although the revised bottom-up estimate of 25 Gg yr$^{-1}$ mentioned above contributes considerably to closing the gap between bottom-up and top-down emission estimates, this new bottom-up value is still lower than the average SPARC-merged top-down emission estimate of $35 \pm 16$ Gg yr$^{-1}$ (though the uncertainty is large). The discrepancy between bottom-up and top-down emission estimates implies the existence of unidentified sources and/or unreported industrial emissions.

Regional studies of episodic enhancements of $CCl_4$ above atmospheric background concentrations observed in several regions

using inverse model techniques, have suggested emissive fluxes of $0.11 \pm 0.04$ Gg yr$^{-1}$ in 2009–2012 from Australia (Fraser et al., 2014), 15 (10–22) Gg yr$^{-1}$ in 2007 from East Asia (Vollmer et al., 2009), 4 (2–6.5) Gg yr$^{-1}$ in 2008–2012 from the U.S. (Hu et al., 2016), and $2.3 \pm 0.8$ Gg yr$^{-1}$ in 2006–2014 from Western Europe (Graziosi et al., 2016). The summed emissions were estimated to total $21 \pm 8$ Gg yr$^{-1}$ (Liang et al., 2016), with the most significant contribution belonging to East Asia. As the sum of regional emissions quantified to date has not accounted for global top-down emissions, an improved quantification of regional/country-

scale and industry-based $CCl_4$ emissions is required to gain a better insight into the causes of the discrepancy between the regional sums and the global top-down estimate. This would improve our understanding of the unidentified and/or unreported industrial emission sources and would help to establish practical and effective regulation strategies.

With the aim of resolving the apparent $CCl_4$ budget discrepancy, this study presents an estimate of regional $CCl_4$ emissions from China, one of the MP Article 5 countries in East Asia. Due to its recent and ongoing strong industrial growth, current emissions

and changes in emission patterns are of special interest. In addition, recent studies based on atmospheric monitoring have consistently reported a significant increase in the emissions of most halocarbons in China (Vollmer et al., 2009, Kim et al., 2010, Li et al., 2011). Top-down estimates of Chinese emissions for $CCl_4$ have been made in previous studies using a Lagrangian inverse model based on ground-based monitoring data (Vollmer et al., 2009) and an interspecies correlation method based on aircraft observations (Palmer et al., 2003; Wang et al., 2014). The estimates made in these studies were quite variable with $17.6 \pm 4.4$ Gg

yr$^{-1}$ in 2001 (Palmer et al., 2003), 15 (10–22) Gg yr$^{-1}$ in 2007 (Vollmer et al., 2009) and $4.4 \pm 3.4$ Gg yr$^{-1}$ in 2010 (Wang et al., 2014), and these studies were conducted prior to the complete phase-out of $CCl_4$ production for emissive applications in China came into effect in 2010. Most recently, Bie et al. (2017) published post-2010 bottom-up emission estimates for China of 4.3 (1.9–8.0) Gg yr$^{-1}$ in 2011 and 5.2 (2.4–8.8) Gg yr$^{-1}$ in 2014, which updated the previous zero emissions estimate (Wan et al., 2009) by including the conversion of $C_2Cl_4$ emissions to $CCl_4$ as well as the source of $CCl_4$ from coal combustion smog.

In this study, we present an 8-year record of continuous, high frequency, high-precision, atmospheric $CCl_4$ concentrations measured at the Gosan station (33° N, 126° E) on Jeju Island, Korea for 2008–2015. Using a tracer-tracer correlation method (Li et al., 2011) based on a top-down interpretation of atmospheric observations, we estimate yearly emission rates of $CCl_4$ for China and examine changes in these rates following the scheduled phase-out for $CCl_4$ in 2010. Gosan station monitors air masses arriving from a variety of different regions (Kim et al., 2012), and the emission footprints of these cover an area from north-eastern China down

to south of the Yangtze River, which is the most industrialized region in China. We also analyze the measurements of 17 other

anthropogenic compounds to identify key industrial sources of CCl$_4$ emissions and their potential locations using a Positive Matrix Factorization model in combination with trajectory statistics (Li et al., 2014).

## 2. Data overview

2.1. Measurements of CCl$_4$ at Gosan

Gosan station (GSN) is located on the remote south-western tip of Jeju Island, which lies to the south of the Korean peninsula (72 m above sea level), and is well situated for monitoring long-range air mass transport from surrounding regions (Fig. S1). Wind patterns at GSN are typical of the Asian Monsoon, with strong predominant north-westerly and north-easterly continental outflows of polluted air from fall through to spring, clean continental air flowing directly from northern Siberia in winter, and pristine maritime air from the Pacific in summer (Fig. S2). High-precision and high-frequency measurements of 40 halogenated compounds

including CCl$_4$ were made continuously every two hours from 2008 to 2015 using a gas chromatography-mass spectrometer (GC-MS) coupled with an online cryogenic pre-concentration system ("Medusa") (Miller et al., 2008) as part of the Advanced Global Atmospheric Gases Experiment (AGAGE) program. Precisions (1$\sigma$) derived from repeated analysis ($n = 12$) of a working standard of ambient air were better than 1 % of background atmospheric concentrations for all compounds, e.g. $\pm$ 0.8 ppt (1$\sigma$) for 85.2 ppt of CCl$_4$. The measurements are mostly on calibration scales developed at the Scripps Institution of Oceanography (SIO).

2.2 Results

The 8-year observational record of CCl$_4$ analyzed in this study is shown in Fig. 1. It is apparent that pollution events (red dots) with significant enhancements above "background" levels (black dots) occurred frequently, resulting in daily variations of observed concentrations with relative standard deviations (RSDs) of 4−20% (in contrast to the RSDs of 0.1−1.5% shown in all the remote

stations operated under the AGAGE program). These results clearly imply that CCl$_4$ emissions are emanating from East Asia. The background concentrations at GSN were determined using the statistical method detailed in O'Doherty et al. (2001), and they agree well with those observed at the Mace Head station (53°N, 10°W) in Ireland (which is representative of a remote background monitoring station in the Northern Hemisphere) and are declining at a similar rate to the global trend (Fig. S4). The magnitude of pollution data analyzed in this study was defined as the observed enhancements (red dots in Fig. 1) in concentration units above

the baseline values (i.e., background values representing regional clean conditions without regional/local pollution events, black dots), to exclude the influence of trends and/or variability in background levels from the analysis.

## 3. Potential source regions of CCl$_4$ in East Asia

A statistical analysis combining enhanced concentrations (above-baseline concentrations) of CCl$_4$ from 2008 to 2015, with

corresponding back trajectories, enabled identification of the regional distribution of potential CCl$_4$ emission sources. The statistical method (see "Trajectory Statistics" in SI) was first introduced in 1994 (Seibert et al., 2004) and has previously been applied to analyses relating to halogenated compounds (e.g., Li et al., 2014; Reimann et al., 2004).

An elevated concentration at an observation site is proportionally related to both the average concentration in each grid cell over which the corresponding air mass has travelled and the air mass trajectory residence time in the grid cell. This allows the method

to compute a residence-time-weighted mean concentration for each grid cell by simply superimposing the back trajectory domain on the grid matrix. We used 6-day kinematic backward trajectories arriving at a 500 m altitude above the measurement site that were calculated using the HYbrid Single Particle Lagrangian Integrated Trajectory (HYSPLIT) model of the NOAA Air Resources Laboratory (ARL) based on meteorological information from the Global Data Assimilation System (GDAS) model with a 1°×1°

grid cell (Li et al., 2014). The residence times were calculated using the methods of Poirot and Wishinski (1986). To eliminate low confidence level areas, we applied a point filter that removed grid cells that had less than 12 overpassing trajectories (Reimann et al., 2004).

The resulting map of potential source areas for $CCl_4$ in East Asia (Fig. 2) shows that emission sources are widely distributed in China, but they are particularly concentrated in north eastern China and south-central China (approximately Shandong, Henan, Hubei, and Guangdong provinces). These provinces include industrialized urban areas that conduct intensive industrial activities, such as chemical manufacturing (http://eng.chinaiol.com/). It is of note that this statistical analysis has little sensitivity to emissions from southwest China, due to the limits of the typical 5- to 6-day back-trajectory domain of the HYSPLIT model. Additionally, this method tends to underestimate the inherently sharp spatial gradients in the vicinity of emission hot-spots, because its calculation scheme distributes the measured concentrations evenly throughout grid cells over which a trajectory has passed (Stohl, 1996). Nonetheless, it is clear that the $CCl_4$ emission sources from East Asia were predominantly located in China.

## 4. Using observed interspecies correlations to estimate country-based, top-down $CCl_4$ emissions in China

To identify pollution events influenced solely related toy Chinese emissions, we classified an event as "Chinese" if the 6-day kinematic back trajectories arriving at GSN had entered the boundary layer (as defined by HYSPLIT) only within the Chinese domain, which was defined as a regional grid of 100–124°E and 21–45°N (Fig. S5(a)). This analysis classified 29% of all observed $CCl_4$ pollution events from 2008 to 2015 (Fig. S5(b)) as "Chinese". An additional 46% were affected by Chinese domain plus another country's; however, these blended air masses were excluded from the determination of Chinese emissions.

For the Chinese emissions estimate of $CCl_4$, we use an interspecies correlation method, analogously to many recent emission studies (e.g., Kim et al., 2010; Li et al., 2011; Palmer at al., 2003; Wang et al., 2014). In this method, the emission rate of a co-measured compound of interest can be inferred based on its compact empirical correlation with a reference compound whose country-scale emission has been independently well-defined. This empirical ratio approach provides a simple yet comprehensive method for estimating regional emissions of almost all halogenated compounds measured at GSN, and it minimizes the uncertainties inherent in more complex modeling schemes. This method is particularly useful for compounds such as $CCl_4$, where the associated bottom-up inventories indicate close to zero emissions and/or clearly have large errors, which thus makes it difficult to adequately define the prior emissions required for inverse modeling. However, the ratio method is restricted by its core assumptions: that the emissions of the reference and target compounds are co-located (or at least well mixed) until they reach the measurement site, and that the reference emissions are well-known. The interspecies ratios we observed at GSN showed statistically significant correlations for many compounds at national scales (Li et al., 2011), suggesting that overall these core assumptions were satisfied in this study.

An adequate reference compound should be a widely used industrial species with high national emission rates, thereby allowing for robust and compact correlations with many other species and low uncertainties in its own emission estimate. The reference compound was chosen by examining the observed relationships of $CCl_4$ enhancements above baseline versus the enhancements above baseline for 25 other halocarbons in air masses classified as "Chinese". We found that the $\Delta CCl_4/\Delta HCFC\text{-}22$ ratio (0.13 ppt/ppt) showed one of the most significant correlations ($R^2 = 0.72$, $p < 0.01$) (Fig. S6). Furthermore, given that China has been the largest producer and consumer of HCFCs since 2003, and that production of HCFC-22 accounts for more than 80% of all Chinese HCFC production (UNEP, 2009), HCFC-22 is the best-suited reference compound for use with China. Additionally, strong Chinese HCFC-22 emissions have been determined from atmospheric observations and inverse modeling in previous studies (Kim et al., 2010; Li et al., 2011; Stohl et al., 2010; An et al., 2012; Fang et al., 2012), with estimates ranging from 46 to 146 Gg yr$^{-1}$

over the period 2007–2009. Our estimates of annual HCFC-22 emissions in China for 2008–2015 were independently derived from atmospheric measurements at GSN using an inverse technique based on FLEXible PARTicle dispersion model (FLEXPART) Lagrangian transport model analysis (Stohl et al., 2010; Fang et al., 2014), and ranged from 89 Gg yr$^{-1}$ in 2011 to 144 Gg yr$^{-1}$ in 2015. The uncertainty in the top-down estimates was 30%, which mainly related to an assumed uncertainty of ±50% in annual prior emissions used for the inversion calculation (Fig. S7).

Next, we used empirical correlations between observed enhancements of $CCl_4$ and HCFC-22 ($\Delta CCl_4/\Delta$HCFC-22; annual slopes shown in Fig. S8) to estimate $CCl_4$ emission rates. The interspecies slopes were determined based on observed enhancements obtained by subtracting regional background values from the original observations, to avoid potential underestimation of the slopes due to the high density of low background values (following Palmer et al. (2003)). Estimated uncertainties for our $CCl_4$ emission estimates comprise the emissions uncertainty of HCFC-22 and an uncertainty associated with the $\Delta$HCFC-22/$\Delta CCl_4$ slope, which was calculated using the Williamson-York linear least-squares fitting method (Cantrell, 2008), considering measurement errors of both HCFC-22 and $CCl_4$.

Figure 3 provides the annual $CCl_4$ emissions in China for the years 2008–2015, which were calculated based on our interspecies correlation method, and also shows a comparison between our results and previous estimates of $CCl_4$ emissions from China. The $CCl_4$ emission rate of 16.8 ± 5.6 Gg yr$^{-1}$ in 2008 found in this study is consistent with 2001 (Palmer et al., 2003) and 2007 (Vollmer et al., 2009) top-down emissions estimates of 17.6 ± 4.4 Gg yr$^{-1}$ and 15 (10–22) Gg yr$^{-1}$, respectively. To obtain those results, Palmer et al. (2003) used observed correlations of $CCl_4$ with CO as a tracer to investigate $CCl_4$ emissions in aircraft observations of the Asian plume over a two-month period (March to April) in 2001, and Vollmer et al. (2009) estimated the 2007 emissions using an inverse model based on atmospheric measurements taken from late 2006 to early 2008 at an inland station (Shangdianzi, 40°N, 117°E) located in the North China Plain. Wang et al. (2014) obtained aircraft measurements over the Shandong Peninsula on July 22 and October 27 in 2010 and from March to May in 2011, and estimated $CCl_4$ emission in 2010 based on observed correlations of $CCl_4$ with both CO and HCFC-22. However, the estimates from these two different tracers differed by ~100 % (8.8 versus 4.4 Gg yr$^{-1}$) and were much lower than the two previous results of Palmer et al. (2003) and Vollmer et al. (2009) and our 2010 estimate of 32.7 ± 5.1 Gg yr$^{-1}$. Although the cause of this discrepancy is unclear, it is considered that it could be related to the low numbers of observations obtained in the aircraft campaigns and to difficulties defining regional background values and extracting pollution signals from the aircraft data. It is also possible that the results mostly represent emissions from northern China. Extrapolating to the entire country using data from northern China would lead to an underestimate of emissions, as most industrial activities occur in the south-central and eastern parts of China.

Our estimates show that Chinese emissions increased sharply before reaching a maximum in 2009–2010 (with a range of 38.2 ± 5.5 to 32.7 ± 5.1 Gg yr$^{-1}$) immediately prior to the scheduled phase-out of $CCl_4$ by 2010. The sudden large increase could be attributed to uncontrolled use/production leading to emissions of stored $CCl_4$ before the scheduled restrictions came into effect. Interestingly, this increase in our emission estimates was also consistent with the increase of about 20 Gg yr$^{-1}$ in the total annual production of $CCl_4$ in China from 2008 to 2010, which was mainly related to an increase in the feedstock production sector, i.e., raw material production for non-ODS chemicals (Bie et al., 2017). After a dip in 2012, our estimated emissions in 2013–2015 remain stable and are similar overall to those in 2011, with no statistically discernible differences between these years. It is of note that the average emission rate estimated in this study of 23.6 ± 7.1 Gg yr$^{-1}$ for the years 2011–2015 is significant, as post-2010 bottom-up emissions of $CCl_4$ in China have been reported as near zero (Wan et al., 2009), and even the most up-to-date bottom-up estimates (Bie et al., 2017) have indicated emissions of only 4.3 (1.9–8.0) Gg yr$^{-1}$ in 2011 and 5.2 (2.4–8.8) Gg yr$^{-1}$ in 2014. These discrepancies between bottom-up and top-down emission estimates may suggest that emissions of $CCl_4$ from either non-

regulated feedstock/process agent use, or unreported non-feedstock emissions from the production of chloromethanes ($CH_3Cl$, $CH_2Cl_2$, $CHCl_3$) and PCE, are larger than expected.

## 5. Industrial source apportionment of atmospheric $CCl_4$ in East Asia

The Positive Matrix Factorization (PMF) Model was used to characterize key industrial $CCl_4$ sources based solely on atmospheric observations (Paatero and Tapper, 1994). We included all $CCl_4$ enhancement events observed at GSN thereby representing better characterization of emission sources throughout East Asia and not just in China. The PMF model has been widely used to identify and apportion sources of atmospheric pollutants (Guo et al., 2009; Lanz et al., 2009, Li et al., 2009; Choi et al., 2010), and is an optimization method that uses a weighted least squares regression to obtain a best fit to the measured concentration enhancements of chemical species (details in SI text) and to resolve the number of "source factors" controlling the observations. A brief mathematical expression of the model is given by Eq. (1),

$$x_{ik} = \sum_{j=1}^{p} g_{ij}f_{jk} + e_{ik} (i = 1,2, \ldots, m; j = 1,2, \ldots, p; k = 1,2, \ldots, n), \tag{1}$$

where $x_{ik}$ represents enhanced concentrations in the time series of the $i^{th}$ compound at the $k^{th}$ sampling time; $g_{ij}$ is the concentration fraction of the $i^{th}$ compound from the $j^{th}$ source; $f_{jk}$ is the enhanced concentration from the $j^{th}$ source contributing to the observation at the $k^{th}$ time, which is given in ppt; $e_{ik}$ is the model residual for the $i^{th}$ compound concentration measured in the $k^{th}$ sampling time; and $p$ is the total number of independent sources (i.e., the number of factors) (Paatero and Tapper, 1994). The number of source factors is an optimal value determined based on the R-squared that measures how close the predicted concentrations are to the observed enhancements of 18 species (including not only $CCl_4$, major CFCs, HCFCs, HFCs, PFCs, $SF_6$, carbonyl sulfide (COS), but also $CH_3Cl$, $CH_2Cl_2$, $CHCl_3$, and PCE) to account for the potential chemical intermediate release of $CCl_4$ during industrial activities. The model's R-squared values, as estimated from a correlation plot between the measured and PMF model-predicted concentrations, showed that an eight–source model is most appropriate, suggesting eight potential source categories for those 18 species. Each source factor is defined based on the source profile (i.e., relative abundances of individual species). The percentage contributions of factors to the observed enhancements of individual compounds are shown in Fig 4. Uncertainties were determined from the 1σ standard deviation of factor contributions from 5 sets of 20 runs (total 100 replications) (Reff et al., 2007).

Factor (A) shown in the Fig. 4 is characterized by $38 \pm 4\%$ of $CCl_4$ and $97 \pm 2\%$ of $CH_3Cl$, suggesting advertent or inadvertent co-production and escape of $CCl_4$ during chloromethane generation in chemical plants (see SI text for chemical reactions). $CCl_4$ and $CH_3Cl$ co-emitted in smog from coal combustion (Li et al., 2003) are less likely to be the source of this factor because COS, which is a major coal burning tracer, does not contribute to this factor. Source factor (B) is largely related to fugitive emissions in feedstock and process agent use of various compounds; it accounts for a large fraction of $CCl_4$ ($32 \pm 4\ \%$) and shows high percentages for several compounds: $72 \pm 18\%$ of $CH_2Cl_2$, $59 \pm 11\%$ of $CHCl_3$, $39 \pm 10\%$ of CFC-11, and $51 \pm 12\%$ of HFC-23. It is of note that $CH_2Cl_2$ and $CHCl_3$ can be produced as by-products of chlorination along with $CCl_4$ and are used as intermediates or solvents in chemical manufacturing. $CCl_4$ is a feedstock for PCE, HFC, methyl chloride, and divinyl acid chloride production (Liang et al., 2016) and is also used in CFC production (Zang et al., 2010; Sherry et al., 2018). In addition, $CHCl_3$ can be used as a feedstock for HCFC-22 production (Montzka et al., 2011), which is consistent with factor (B) also being distinguished by a high contribution of HFC-23: Chinese emissions of HFC-23 account for ~70% of total global emissions (Kim et al., 2010; Li et al., 2011) and it is a typical by-product of HCFC-22 generation (Fang et al., 2015). HFC-23 is thus emitted at factory level in regions where chemical manufacturing industries are heavily collocated. Overall, the fact that observed enhancements of HFC-23, $CCl_4$, $CH_2Cl_2$, $CHCl_3$, and CFC-11 are grouped together into the factor (B) in the PMF analysis implies that this factor most likely represents fugitive emissions of these compounds occurring at the factory level during various chemical manufacturing processes

in China. Source factor (C) is distinguished by $19 \pm 1\%$ of $CCl_4$ and $95 \pm 2\%$ of PCE; it can possibly be explained by advertent or inadvertent co-production and escape of $CCl_4$ during industrial $C_2Cl_4$ production and in part by fugitive emissions of $CCl_4$ used as a chlorination feedstock for $C_2Cl_4$ production.

The spatial distributions (Fig. S9) of source factors (A), (B), and (C) derived from trajectory statistics (SI text) are similar and cover areas in and around Guangzhou of Guangdong, Wuhan of Hubei, Zhengzhou of Henan, and Xian of Shaanxi province. These distributions are consistent with the results of PMF analysis, which confirms that $CCl_4$ emissions from China are more strongly associated with industrial processes than with population density. Our results are also consistent with those of a previous study on halocarbons observations in the Pearl River Delta region of Guangdong (Zhang et al., 2010), which used a source profile analysis to reveal that CFCs and $CCl_4$ emissions from an industrial source related to chemical (i.e., refrigerant) production, increased by 1.4–2.0 times from 2001−2002 to 2007, even though there were no significant changes in the atmospheric mixing ratios of these compounds for the 6 years. These results imply the increased use of $CCl_4$ in chemical production. The three emission source factors (A−C), which account for $89 \pm 6\%$ of $CCl_4$ enhancements observed at GSN, are thus considered to be mostly escaped $CCl_4$ emissions at the factory level relating to inadvertent by-production, feedstock usage for production of chlorinated compounds, and process agent use for chemical processes.

Other factors of PMF analysis relate to (D) primary aluminum production (Blake et al., 2004), (E) HFCs production/applications, (F) refrigerant consumption, (G) processes in the semiconductor/electronics industry, and (H) foam blowing agent use, and can mostly be summarized as being distributed emissions. However, the percentage contributions of these other source factors to $CCl_4$ enhancements are not statistically significant when considering the uncertainty range. The smallest contribution to $CCl_4$ of the sources characterized as general consumption and legacy release could suggest that $CCl_4$ emissions from old landfills, contaminated soil, and solvent usage have become less significant. A detailed description of factors D−H is provided in the Supporting Information section.

## 6. Conclusions

An 8-year record of atmospheric $CCl_4$ observations obtained at GSN provided evidence of ongoing $CCl_4$ emissions from East Asia during 2008–2015. Based on these measurements, this paper presents a top-down $CCl_4$ emissions estimate from China of $23.6 \pm 7.1$ Gg $yr^{-1}$ for the years 2011–2015, which is different to a bottom-up estimate of 4.3–5.2 Gg $yr^{-1}$ given by most current bottom-up emission inventories for post-2010 China.

Liang et al. (2016) estimated global top-down emissions as $35 \pm 16$ Gg $yr^{-1}$, which was an average estimate based on the estimate of $40 \pm 15$ Gg $yr^{-1}$ for the new 33-year total lifetime of $CCl_4$ and an independent top-down method using the observed inter-hemispheric gradient in atmospheric concentrations which yielded $30 \pm 5$ Gg $yr^{-1}$. The SPARC sum of regional emissions was estimated as $21 \pm 8$ Gg $yr^{-1}$, of which Chinese emissions of 15 (10–22) Gg $yr^{-1}$ contributed $71 \pm 33\%$ to the total amount, but this result is still lower than the aggregated top-down values. However, if we employ the higher emission estimate of $23.6 \pm 7.1$ Gg $yr^{-1}$ obtained for China in this study, the summed regional estimate would be $30 \pm 10$ Gg $yr^{-1}$, which is largely in agreement with the best global emissions estimate of $35 \pm 16$ Gg $yr^{-1}$ determined by Liang et al. (2016).

A factor analysis combining the observed concentration enhancements of 18 species was used to identify key industrial sources for $CCl_4$ emissions and to link our atmospheric observation-based top-down identification of potential sources with bottom-up inventory-based estimates (e.g., Liang et al., 2016; Sherry et al., 2017). Three major source categories accounting for $89 \pm 6\%$ of $CCl_4$ enhancements observed at GSN were identified as being related to advertent or inadvertent co-production and escape of $CCl_4$ from $CH_3Cl$ production plants (factor (A)),escape during industrial PCE production (factor (C)), and fugitive emissions (factor

(B)) from feedstock use for the production of other chlorinated compounds (e.g., $CHCl_3$) and process agent use, and possibly from other chloromethanes use in chemical manufacturing. These sources are largely consistent with the bottom-up $CCl_4$ emissions pathways identified in SPARC (Liang et al., 2016). The SPARC estimate of global $CCl_4$ emissions from chloromethanes and PCE/$CCl_4$ plants (pathway B from Liang et al. (2016) and Sherry et al. (2018)) was 13 Gg yr$^{-1}$, as the most significant source.

Fugitive feedstock/process agent emissions, denoted as pathway A by Liang et al. (2016) and Sherry et al. (2018), were estimated as ~2 Gg yr$^{-1}$. The emissions contributions from China to pathways B and A were 6.6 Gg yr$^{-1}$ and 0.7 Gg yr$^{-1}$, respectively (Liang et al., 2016; Sherry et al., 2018).

If we assume that emission rates from sources correspond to the relative contributions of corresponding source factors to the total Chinese emission rate ($23.6 \pm 7.1$ Gg yr$^{-1}$ for the years 2011–2015), source factors (A) ($CCl_4$ emissions from chloromethane plants)

and (C) (emissions from PCE plants) amount to $13 \pm 4$ Gg yr$^{-1}$ for China. This is as high as the global bottom-up number of 13 Gg yr$^{-1}$ for pathway B emissions and more than 50% higher than the Chinese estimate of 6.6 Gg yr$^{-1}$. This could represent that the ratio of $CCl_4$ emissions from these processes into the atmosphere may be higher than previously assumed, although factor (C) could possibly include the influence of fugitive emissions of $CCl_4$ when using as a chlorination feedstock for PCE production. Furthermore, source factor (B) (fugitive feedstock/process agent emissions) are estimated at $~7 \pm 2$ Gg yr$^{-1}$ from China alone,

which again contrasts with the Chinese estimate of ~0.7 Gg yr$^{-1}$ and even with the lower global estimate of only 2 Gg yr$^{-1}$ for pathway A from Liang et al. (2016) and Sherry et al. (2018). Although the analysis provided here may contain uncertainties, it appears that the SPARC industry-based bottom-up emissions are underestimated. Therefore, improvements in estimating industry bottom-up emissions of $CCl_4$, particularly at the factory and/or process level, are crucial for gaining a better understanding and evaluation of ongoing global emissions of $CCl_4$.


Data used in this study are available from http://agage.eas.gatech.edu/data_archive/agage/gc-ms-medusa/.

Acknowledgments: This research was supported by the National Strategic Project-Fine particle of the National Research Foundation of Korea (NRF) funded by the Ministry of Science and ICT (MSIT), the Ministry of Environment (ME), and the

Ministry of Health and Welfare (MOHW) (No. NRF-2017M3D8A1092225). We acknowledge the support of our colleagues from the Advanced Global Atmospheric Gases Experiment (AGAGE). The operation of the Mace Head AGAGE station, the MIT theory and inverse modeling and SIO calibration activities are supported by the National Aeronautics and Space Administration (NASA, USA) (grants NAG5-12669, NNX07AE89G, NNX11AF17G and NNX16AC98G to MIT; grants NAG5-4023, NNX07AE87G, NNX07AF09G, NNX11AF15G, NNX11AF16G, NNX16AC96G and NNX16AC97G to SIO). AGAGE stations operated by the

University of Bristol were funded by the UK Department of Business, Energy and Industrial Strategy (formerly the Department of Energy and Climate Change) through contract TRN 34/08/2010, NASA contract NNX16AC98G through MIT, and NOAA contract RA-133R-15-CN-0008.

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

Figure Captions

**Figure 1.** Atmospheric $CCl_4$ concentrations observed from 2008 to 2015 at Gosan station (GSN, 33°N, 126°E) on Jeju Island, Korea. Pollution events (identified as significant enhancements in concentrations from background levels shown in black) are denoted by red dots.

5 **Figure 2**. Distribution of potential source regions calculated from trajectory statistics for enhancement data of $CCl_4$ observed from 2008 to 2015. The color code (in ppt) denotes a residence-time-weighted mean concentration for each grid cell. The resulting map of potential source areas for $CCl_4$ shows that emission sources are widely distributed over China. The site of Gosan station is indicated by an asterisk (*).

**Figure 3.** $CCl_4$ emissions in China as determined by an inter-species correlation method. A comparison between our results and 10 previous estimates for Chinese emissions is also shown. Note that emissions reached a maximum in 2009−2010 in concurrence with the scheduled phase-out of $CCl_4$ by 2010, but the average annual emission rate of $23.6 \pm 7.1$ Gg yr$^{-1}$ for the years 2011−2015 are still substantial.

**Figure 4.** Source profiles derived from PMF analysis for 18 compounds, including $CCl_4$, CFCs, HCFCs, HFCs, PFCs, $SF_6$, COS, 15 $CH_3Cl$, $CH_2Cl_2$, $CHCl_3$, and $C_2Cl_4$. The PMF analysis is performed on the time series of enhanced concentrations. The y-axis shows the percentage of all observed enhancements associated with each factor (with 1σ standard deviation) such that the vertical sum for each species listed on the x-axis is 100.

Figure 1.

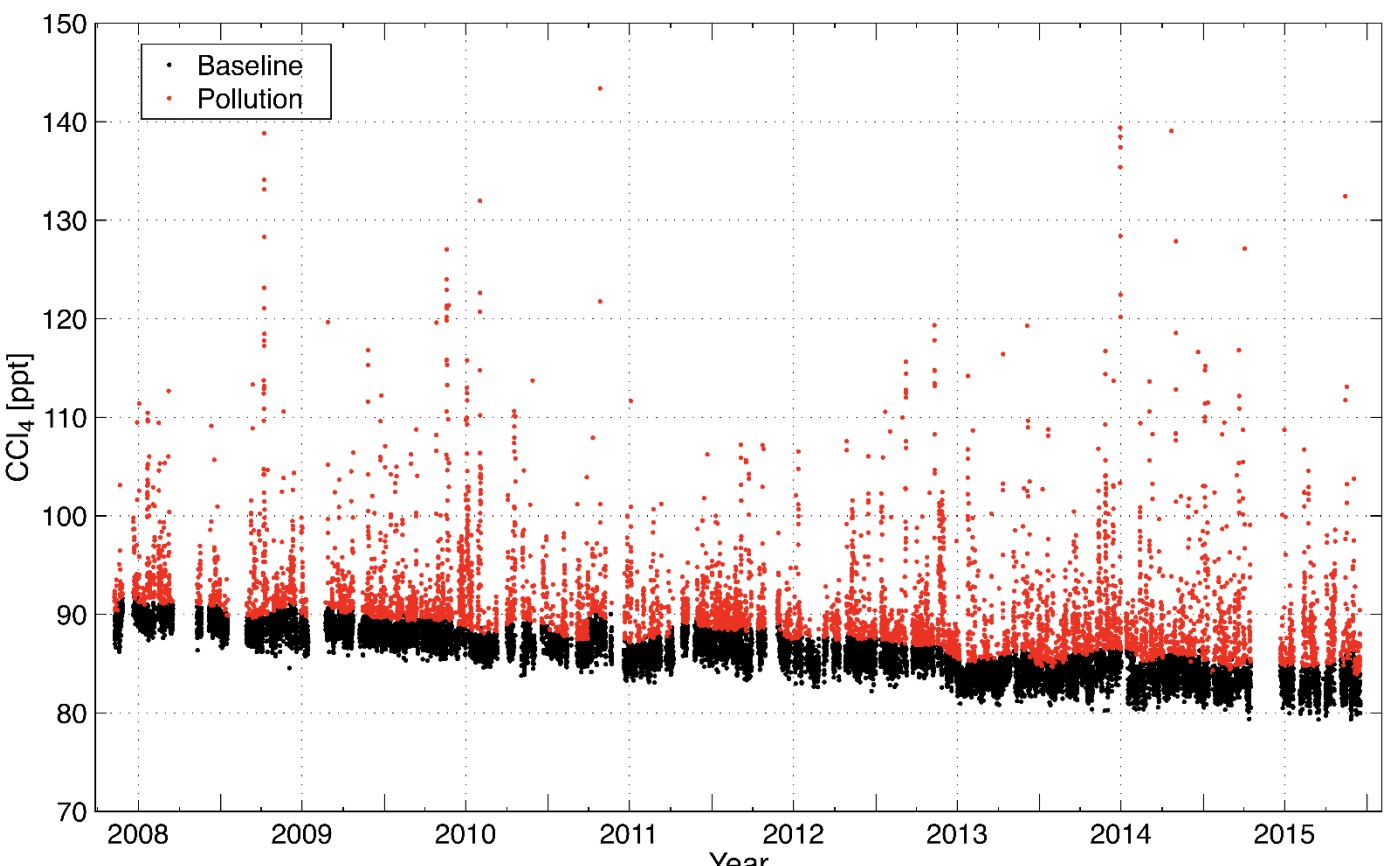

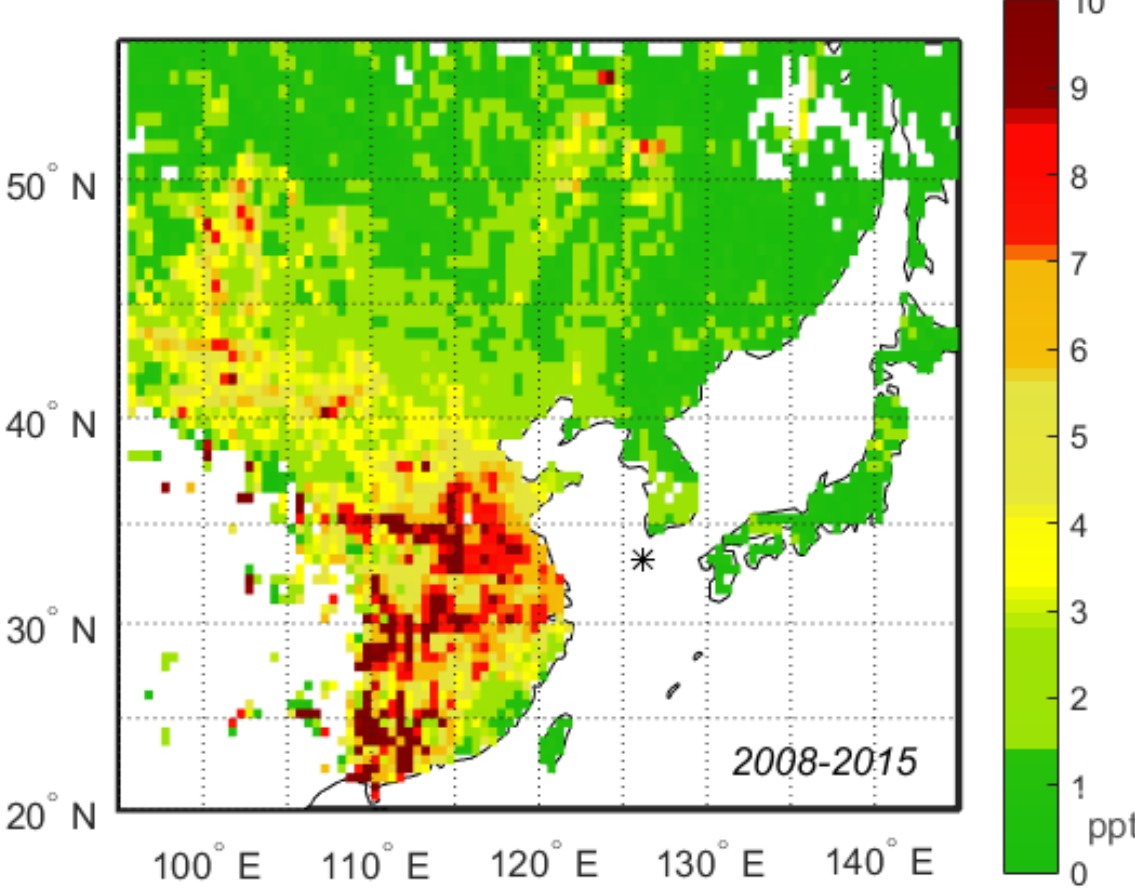

Figure 2.

Figure 3.

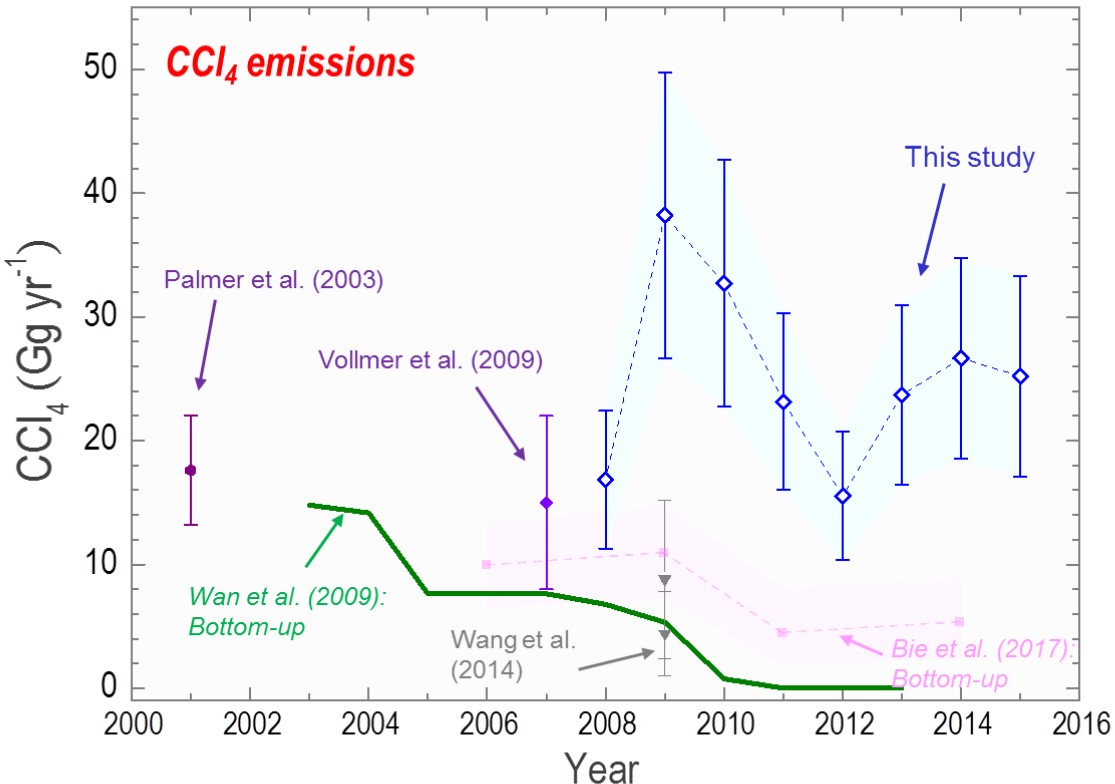

Figure 4.

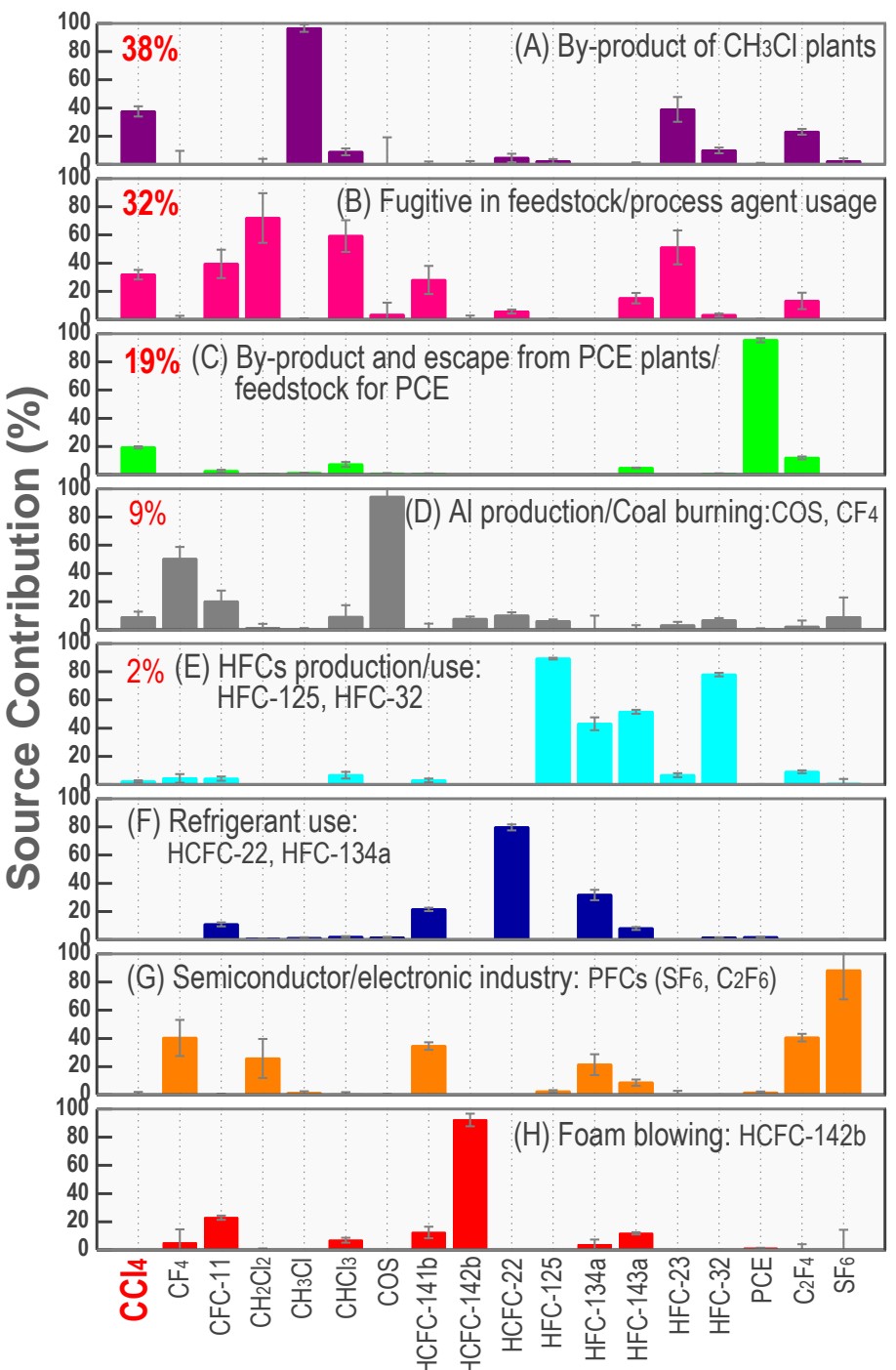