# Peer review of "Supporting Information"

_Atmospheric Chemistry and Physics, 2018_

## Referee Comment (RC1) · Anonymous Referee #1 · 16 Mar 2018

Paper Summary: This paper uses observations from the Korean Gosan station to identify the location of CCl4 sources and the specific industrial processes involved with the CCl4 emissions. There are two basic techniques used to analyzed these data. First, trajectories are used in a source/receptor analysis technique to identify the CCl4 emission locations. The major sources originate in Eastern China. Second, a positive matrix factorization (PMF) analysis technique is used to finger-print the specific sources of CCl4. This analysis reveals that the emissions are primarily from chloromethane production (CH3Cl), perchloroethylene production, and fugitive emissions from feedstock usages. These estimates are larger than those from the SPARC CCl4 report, with the fugitive emissions being 10x larger than SPARC!
Review Summary: This is an excellent paper that NEEDS to be published. My overall comments are with regard to improving the writing in the paper and some of the structure.

Paper Suggestions:

While the paper mentions the SPARC (2016) CCl4 report, there ought to be more discussion of how this paper seems to resolve the discrepancy between their emissions based upon measurements. SPARC had a top-down emissions calculation of 40±15 Gg/y, a hemispheric gradient method of 30±5 Gg/y, and a regional emission estimates of 21.4±7.5 Gg/y. The SPARC regional 21.4 Gg/y had a 15 Gg/y contribution from China. The higher estimate herein of 24 Gg/y from China would bring this 21.4 SPARC number up to 30 Gg/y - in precise agreement with the gradient method and within the uncertainties of the 40 Gg/y top down estimate.

The discussion in the summary of the CCl4 sources should be broken out with more definitive statements. The SPARC report used industrial estimates to characterize potential sources [Sherry et al., 2016], and this paper provides the **first observational basis** for these sources, but this paper also makes the case that Sherry et al. is perhaps too conservative in their estimates.

The paper is fairly well written, but many of the current paragraphs need to be broken up into more distinct sections or primary thoughts. The extended paragraphs of the current version obscure the thoughts, logic of the paper, and the overall content of the text. For example, the 2nd para of the Introduction (P2, 4-30) talks about top down emissions, bottom up emissions, . . .. I would break this up into paras on: 1) top down emissions (4-12); 2) a SPARC bottom up para (12-16); and 3) a discussion of regional emissions.

In the 1st para of section 3 (P. 4 line 18 to P.5 line 32 - 46 lines!), there are a broad range of paragraph thoughts. The paragraph starts with a discuss of the interspecies correlation and ends with a thought on an underestimate of Chinese emissions. Please

break this up to improve the flow of the text.

The "Data Overview" section both discusses the data and shows results. I would re-structure sections 2 and 3 into: a data, methods, and results sections. The Supporting Information ought to flow better into these data and methods sections.

Again, break up the single paragraph of the conclusions into short paragraphs. The main messages are lost in this "run-on" paragraph.

Figures are good. For Fig. 4, put some vertical lines on the plot to see how the bars line up with chemical names at the bottom.

Fig. S5. What are the colors for? Do they indicate statistical significance?

---

## Referee Comment (RC2) · Anonymous Referee #2 · 30 Mar 2018

Park et al presented a top-down emissions estimate of CCl4 from East Asia based on high frequency surface measurements of halocarbons at the Gosan sites. This paper is timely. Results presented in this paper provide crutial pieces of information that closes the CCl4 global budget as well as providing the atmopsheric observational evidence that unported CCl4 emissions during chloromethans and PCE production. However, the writing in many places can use some improvements. I recommend the authors go through the entire manuscript thoroughly to improve the clarity and accuracy. The paper should be published in ACP after the following comments are addressed.

P1 L15, "the 2010" -> "a 2010" 2. P2 L5-7. You should state that the global top-down

emissions are derived based on both the CCl4 lifetimes and the observed global decline rate. 3. P2 L9. THe global emissions number from Liang et al, 2014 was 39Gg/yr, not 30Gg/yr. 4. P2 L11-12. I am not sure why you say "unidentified sources and/or unreported anthropogenic emissions". CCl4 is a predominantly man-made compound, therefore the emissions sources are anthropogenic. 5. In many places, need to change the "," after the references to ";". 6. P2 L27-30. You need to merge these two sentences and present the results from these studies in a less confusing way with a correct referencing style. In the present form, it is hard for the readers to figure out from which studies the 4.3 and 5.2 Gg/yr were from. 7. P2 L30. Change to "8-year continuous high frequency, high precision atmospheric CCl4 concentrations measured . . ." 8. P3 L2. Change "below the " to "to the south of .." 9. P3 L7. I am not sure what do you mean by "well situated to allow monitoring of long-range transport from the surrounding region". Is this because of elevation or it is in remote clean ocean? By surrounding region, what regions are you referring to? China? The Korean Peninsula? Please clarify. 10. P3 L10. Please include the actual values than just say "high-precision and high-frequency" 11. P3 L22. You need to define what do you mean by "baseline values". This is jargon. 12. P4 L6-7. It would be good to add references here. 13. P7 L11. It is interesting that CFC-11 showed up in the source factor. Does this indicate that CFC-11 is also produced in the CM plants? 14. P7. It will be of great value to CCl4 source identification to link the discussions in the source factors to the industrial production, usage, and potential emissions pathway in Sherry et al. (2017). Such a discussion will help to build link from bottom-up inventory-based estimate to atmospheric observation based top-down estimate. 15. Figure 3 and related discussions. (1) I wonder if part of the difference between the Vollmer et al., 2009 and this study is due to the location of Gosan vs. Shandianzi. The location of Gosan captures most of the outflow from the industrial central and south China, where all the CCl4 production industries are located (as suggested by Figure 2), while Shandianzi captures mostly the air influenced by N. China, without much CM production. Should consider add a related discussion on this in the manuscript. (2) The covariance of CFC-11 and CMs (source factor 2) is

very interesting. Does this mean CFC-11 is also an intendended by-product during the industrial process and the recent increase in CFC-11 unreported emissions (Montzka et al. 2018) is to some extent linked to the CCl4 emissions increase in China between 2012-2016?
* * *

---

## Short Comment (SC1) · 5 Apr 2018

In many respects this is an excellent paper that uses high frequency atmospheric measurements that have low and well defined uncertainty to a) estimate the magnitude of carbon tetrachloride emissions in air reaching the Gosan observatory and b) characterise the geographical location of those emissions. However, the authors go several steps too far when they assign physical processes to the sources of releases of CCl4. Showing where the CCl4 is emitted in China is a major contribution to our understanding of the input of CCl4 to the atmosphere but to then assign emissions to particular industrial processes based solely on correlation with emissions of other chemicals is

an over-interpretation of the data presented and contradicts practical evidence used to build bottom-up inventories for both China and the world as a whole. Consequently, the discussion of results and conclusions on pages 7 to 9 needs to be completely rewritten taking into account that the measurements relate to emissions and do not relate to contemporaneous production in either quantity or location.

The SPARC report on CCl4 [SPARC, 2016] and subsequently Sherry et al. [2018] showed the principal sources of emission to be inadvertent and unreported, arising from:

manufacture and use of chlorine, including as a disinfectant;

leakage from historic landfill;

unreported emissions of material produced during the manufacture of legitimate chemical products and then used in emissive applications in contravention of regulations.

Of the 25 Gg/y emissions estimated [SPARC, 2016] only 2 Gg/y were calculated to be fugitive emissions. Actual fugitive emissions, that is emissions from the plant during the production of CCl4 (either as desired product or as by-product) or its use as a chemical feedstock, are better understood than any of the other sources and account for only a few percent of the global emissions of CCl4. There is an important distinction between fugitive emissions, which are (or at least can be) controlled and which are an economic loss to the manufacturer and material that is disposed of "usefully" but which is, in the end, emitted into the atmosphere.

At the root of the problem with the paper is the naive assertion that all emissions arise from chemical production. Among the 26 halocarbons reported in the paper, this is true only for:

HFC-23 (which arises from HCFC-22 production and where 90% of emissions are from the chemical plant itself [Simmonds et al., 2018]) and

that part of CF4 emissions that arise in aluminium production.

For the rest, emissions occur during use of the substance or when equipment containing the substance is scrapped at the end of its life. This is well known, indeed it forms the basis for the national estimates of fluorinated greenhouse gases submitted to UNFCCC, and the authors have no grounds for asserting otherwise.

From the point of view of assigning geographical locations of CCl4 emissions, the important species are methyl chloride, dichloromethane, tetrachloroethene and HFC-23. These form part of the suite of chemicals used by the authors in the analysis of the observation to show how emissions of CCl4 are geographically co-located with enhanced emissions of other species, with the results shown in Figure 4.

Combustion processes in which the fuel contains chloride ion (much of Chinese coal, municipal solid waste and biomass) account for most of the methyl chloride emitted into the atmosphere by human activity and there is a substantial natural source (five times larger than anthropogenic)[Carpenter & Reimann, 2014]. Sherry et al. [2018] show 26 Gg/y of methyl chloride is made in China from CCl4, so the statement in Figure 4, panel 1 that 38% of release is due to methyl chloride production implies a fugitive emission rate of 35% - ludicrous both from economic and public health points of view.

In Figure 4, panel 2, 32% of CCl4 emissions are co-located with 78% of DCM emissions. DCM has a high solvent power for oils and greases and for some polymeric materials. These properties, coupled with its volatility (boiling point 40.1oC) have led to its wide use as an industrial solvent in applications such chemicals and pharmaceuticals production and to a lower extent as food extraction solvent, for metal cleaning and paint removal. It is also a component of special adhesives and has been used in PU foam blowing, in aerosols, paint strippers and as laboratory agent. Many of these uses can result in much of the DCM employed being emitted into the environment (so-called emissive uses). More recently, smaller quantities of DCM have been used as chemical feedstock to produce HFC-32 (CH2F2, difluoromethane) but use as feedstock does not result in significant emission of DCM. The conclusion must be that 32% of the CCl4 emissions found are co-located with the areas where DCM is emitted

from its use, principally as a solvent.

Exactly the same can be said about Figure 4, panel 3. Here the co-location demonstrated is with tetrachloroethene. Like DCM, this is used in many applications as a solvent, particularly in textile and metal cleaning, which is where most of the emissions arise. Compared to the amount that is produced and used, feedstock emissions are small. So, a further 19% of CCl4 emissions are co-located with solvent emissions of tetrachloroethene.

The first two panels of Figure 4 also show that 70% of CCl4 emissions are co-located with 90% of HFC-23 emissions. CCl4 is not directly involved in the chemical processes that make or use HFC-23. The authors present no evidence to attribute the co-location of these emissions and those of DCM, chloroform and CFC-11 to "fugitive emissions..............at the factory level during various chemical manufacturing processes in China" (page 7, lines 20-21). This is an assumption that, as discussed above, has no basis in fact.

Figure S5 shows correlations between observations of the atmospheric concentrations of 26 halocarbons. Of these, some 19 have correlation coefficients versus CCl4 higher than 0.6 (average 0.72). The conclusion that CCl4 is a ubiquitous contaminant of polluted air samples arising from industrial regions of China is obvious. However, the co-locations demonstrated are for emissions of all the substances, not emissions of one against production of the others.

Apparently, in China some 90 Gg/y of CCl4 are co-produced with other chloromethanes [Zhang et al., 2015]: this represents only 3% of the total Chinese production of methyl chloride, dichloromethane and chloroform, nevertheless it is the main production source of CCl4. SPARC [2016] and Sherry et al. [2018] were able to account for 13 Gg/y globally as "unreported" emissions from NON-FEEDSTOCK USE. As mentioned above, feedstock emissions were calculated separately and shown to be 2 Gg/y. The evidence presented by the authors is entirely consistent with a higher figure for

these unreported emissions from China which would come from industrialised areas, as found. There is, however, no evidence to support the assertion that they arise during production of other chemicals. All that has been proved is that their emissions are geographically co-located with emissions of these other chemicals.

The sentence on page 8 lines 31 to 32 "This seems plausible, as evaporative losses of CCl4 during its use as feedstock and/or process agent and from storage reservoirs of factories are easily overlooked and very poorly constrained" is simply wrong. The highest concentration of CCl4 in the air in a storage tank ventilated to the atmosphere is the equilibrium vapour pressure at the temperature of the liquid CCl4 in the tank. In practise the vapour above the liquid is not well mixed, so the vapour pressure represents an absolute maximum. At 20oC, the vapour pressure is such that a full to empty and refill cycle in such a storage tank could release 0.04% of the CCl4. This is well known, easily calculated, and forms the basis of default estimates of fugitive emissions (such as the 0.5% default factor in the IPCC Greenhouse Gas Reporting Guidelines) [IPCC, 2006]. This sentence will become redundant when the discussion of Figure 4 and the conclusions are completely rewritten.

In summary, the authors need to accept the difference between emissions and production and to rewrite pages 7 to 9 recognising that their observations relate only to emissions. It might help if they were to co-opt one of the authors of the SPARC report or the Sherry et al. [2018] paper to help them write robust conclusions about the relationships between CCl4 and other emissions. The present text is wrong in almost all respects.

References

Carpenter L J and Reimann S (2014) Ozone-depleting substances (ODSs) and other gases of interest to the Montreal protocol, Ch 1 of Scientific Assessment of Ozone Depletion: 2014, Global Ozone Research and Monitoring Project—Report No. 55 (Geneva, Switzerland:World Meteorological Organization).

IPCC (Intergovernmental Panel on Climate Change) (2006) IPCC Guidelines for National Greenhouse Gas Inventories Ch 3.10 Fluorochemicals Production, IPCC and IGES, Tsukuba, Japan, 2006

Li, S., Park, M.K., Jo, C.O. Park S. (2017), Emission estimates of methyl chloride from industrial sources in China based on high frequency atmospheric observations, J Atmos Chem 74: 227. https://doi.org/10.1007/s10874-016-9354-4

Simmonds P.G., M. Rigby, A. McCulloch, M.K. Vollmer, S. Henne, J. Mühle, S. O'Doherty, A.J. Manning, P.B. Krummel, P.J. Fraser, D. Young, R.F. Weiss, P.K. Salameh, C.M. Harth, S. Reimann, C.M. Trudinger, L.P. Steele, R.H.J. Wang, D.J. Ivy, R.G. Prinn, B. Mitrevski and D.M. Etheridge (2018), Recent increases in the atmospheric growth rate and emissions of HFC-23 ($CHF_3$) and the link to HCFC-22 ($CHClF_2$) production, Atmos. Chem. Phys., 18, 1–17, 2018, https://doi.org/10.5194/acp-18-1-2018

Sherry D., A. McCulloch, Q. Liang, S. Reimann and P.A. Newman (2018), Current sources of carbon tetrachloride ($CCl_4$) in our atmosphere Environ. Res. Lett. 13 024004

SPARC (2016), SPARC Report on the Mystery of Carbon Tetrachloride. Q. Liang, P.A. Newman, S. Reimann (Eds.), SPARC Report No. 7, WCRP-13/2016.

Zhang L., Yang W., Zhang L., Li X. (2015), Highly chlorinated unintentionally produced persistent organic pollutants generated during the methanol-based production of chlorinated methanes: A case study in China, Chemosphere, 133, 1-5
* * *

---

## Referee Comment (RC3) · Anonymous Referee #3 · 2 May 2018

There has been a long-standing mystery of why the atmospheric concentration of carbon tetrachloride has declined much slower than predicted after its use was banned by the Montreal Protocol. The SPARC (2016) report resolved only part of this mystery by assessing a slightly longer atmospheric lifetime and by increasing estimates of industrial bottom-up emissions. However, a reconciliation of the top-down and bottom-up estimates was not achievable unless the error bars were stretched to their limits.

The present study by Park et al. utilizes high precision measurements of a suite of halocarbons at a background air monitoring station at Gosan, South Korea, to identify the origins of large fugitive emissions of CCl4 and to estimate their overall emission

rates between 2008-2015. The analysis determines that emissions from heavily indus­trialized regions of China can account for roughly 24 +/- 7 Gg/yr CCl4 between 2011 and 2015 instead of the 4-5 Gg/yr reported bottom-up emissions rates. Surprisingly, emission rates do not seem to have declined over this time period. The additional 19 Gg/yr of fugitive emissions from China would account for over half of the global CCl4 emissions, and perhaps be enough to resolve the remaining mystery of carbon tetrachloride. Thus, this paper represents a very important scientific advance indeed.

The atmospheric measurements are of high quality and the method of using back air trajectories combined with empirical correlations with a reference compound (HCFC-22) is supported by an independent derivation of HCFC-22 emissions that agrees with prior estimates. The industrial source apportionment using the Positive Matrix Factor­ization (PMF) model yielded several strong relationships, pointing to multiple sources of CCl4 associated largely with emissions with other compounds. The interpretation is that the fugitive emissions are occurring at the factory level during production of various chlorocarbons. This seems highly plausible, as the production of these com­pounds are co-located, whereas the consumption of these compounds are expected to be more widely distributed.

Overall, the writing and figures are clear, and the methodology maximizes the function­ality of a high quality dataset. I encourage the publication of this important work, with only a few minor edits suggested below.

1. Pg 2, line 6. The relevant soil sink reference is: Rhew & Happell, 2016, not Rhew et al., 2008.

2. Pg 3, line 10. Here it would be helpful to have a reference or more description about the Gosan station. A brief description of the sample intake line, its height and its proximity to other major landscape features would be helpful details.

3. Pg 3, line 20. The authors should specify that the remote background station in the Northern Hemisphere is Mace Head, Ireland. On a related note, it appears that no

other AGAGE station comes anywhere close to the pollution level events that Gosan station experiences. Expressing this, perhaps in a quantitative way (standard deviation?) would add to the argument that the Gosan station is uniquely situated among the network to capture the primary region of fugitive emissions. After seeing the data published online from all the other stations, it seems clear that this is so.

4. Section 4. Although the time periods may differ, it may be useful to compare these results with some ground based measurements within China that are closer to the source regions. For example, prior studies have found very high concentrations of halocarbons in the Pearl River Delta region of China. Zhang et al., (JGR 115, D15309, 2010) measured elevated concentrations in 2007 and report "The high correlation between CCl4 and CFCs suggests that this source was more related to the production than the consumption of refrigerants." How important is the Pearl River Delta region compared to other regions in the present study? It is difficult to assess based on the maps.

5. Section 4. It would be interesting to see if CH3Br adds any clarity to the model – it is not shown in Figure 4 but shows a high correlation to many other compounds (Figure S5), including CCl4. CH3Br is also banned by the Montreal Protocol but has substantial natural as well as anthropogenic sources. As there are no major natural sources of CCl4, the elevated concentrations of CH3Br may be associated with previously unknown anthropogenic sources. It may be outside the scope of this particular paper, but it would be worth investigating if CH3Br is also emitted from CH3Cl production sites.

6. Pg 9, line 4. The data repository for the Gosan dataset will need to be updated, as the website specified does not appear to have accessible data repositories.

7. Figure S1. The Gosan station should be highlighted with a larger symbol. Also: the dark blue obscures the text and border slightly.

8. Figure S5: The color scheme helps, but the text is very hard to read. Please make the graphic large enough such that the numbers are readable. It appears that the image

can potentially be increased 25% in size while still fitting in the margins. Subscripts can also be added to the left side labels.

9. Figure S7. Why is 2010 in bold and red?

10. Figure S8. What do the colors of the legend indicate?

---

## Author Comment (AC1) · 19 Jun 2018

Referees' comments on "Toward resolving the budget discrepancy of ozone-depleting CCl4: An analysis of top-down emissions from China" by Sunyoung Park, Shanlan Li, Jens Mühle, Simon O'Doherty, Ray F. Weiss, Xuekun Fang, Stefan Reimann, Ronald G. Prinn

We thank the referees for their thoughtful and thorough reviews. We are pleased that all the reviewers see our manuscript as a valuable contribution to the field. We have made changes to the manuscript to answer the suggestions of the reviewers and clarified a few points raised in review. We respond to the referee's comments below and a revised

[Figure]

version of the manuscript including most of the changes suggested by the reviewers will be submitted to the editor. We thank the reviewers and the editor for their time and effort and appreciate the recommendation for publication in Atmospheric Physics and Chemistry.

Reviewer comments:

Referee #1:

Paper Summary: This paper uses observations from the Korean Gosan station to identify the location of CCl4 sources and the specific industrial processes involved with the CCl4 emissions. There are two basic techniques used to analyze these data. First, trajectories are used in a source/receptor analysis technique to identify the CCl4 emission locations. The major sources originate in Eastern China. Second, a positive matrix factorization (PMF) analysis technique is used to finger-print the specific sources of CCl4. This analysis reveals that the emissions are primarily from chloromethane production (CH3Cl), perchloroethylene production, and fugitive emissions from feedstock usages. These estimates are larger than those from the SPARC CCl4 report, with the fugitive emissions being 10x larger than SPARC!

Review Summary: This is an excellent paper that NEEDS to be published. My overall comments are with regard to improving the writing in the paper and some of the structure.

Paper Suggestions: While the paper mentions the SPARC (2016) CCl4 report, there ought to be more discussion of how this paper seems to resolve the discrepancy between their emissions based upon measurements. SPARC had a top-down emissions calculation of 40±15 Gg/y, a hemispheric gradient method of 30±5 Gg/y, and a regional emission estimates of 21.4±7.5 Gg/y. The SPARC regional 21.4 Gg/y had a 15 Gg/y contribution from China. The higher estimate herein of 24 Gg/y from China would bring this 21.4 SPARC number up to 30 Gg/y - in precise agreement with the gradient method and within the uncertainties of the 40 Gg/y top down estimate.

»> Yes, we agree with the reviewer. This is an important point to mention and we have included the following sentences in the second paragraph of the Conclusions section: "According to Liang et al. (2016), global top-down emissions were estimated to be 35 ±16 Gg yr-1 as an average estimate between 40±15 Gg yr-1 based on the new 33-year total lifetime of CCl4, and an independent top-down method using the observed inter-hemispheric gradient in atmospheric concentrations which yielded 30±5 Gg yr-1. The SPARC sum of the regional emissions estimates of 21±8 Gg yr-1, where Chinese emission of 15 (10–22) Gg yr-1 contributed 71±33%, was still less than the aggregated top-down values. Instead, if we apply the higher emission estimate of 23.6±7.1 Gg yr-1 from China suggested here, the summed regional estimate would be calculated to be 30±10 Gg yr-1, largely in agreement with the best estimate of global emissions of 35 ±16 Gg yr-1 from Liang et al. (2016)."

The discussion in the summary of the CCl4 sources should be broken out with more definitive statements. The SPARC report used industrial estimates to characterize potential sources [Sherry et al., 2016], and this paper provides the **first observational basis** for these sources, but this paper also makes the case that Sherry et al. is perhaps too conservative in their estimates.

»> This comment is also very helpful. The factors were re-named as (A), (B), (C) and so on, and their descriptions in the section of "Industrial source apportionment of atmospheric CCl4 in East Asia" were also updated to make it easier to compare them to the SPARC report, as suggested by reviewers 1 and 2. The figure legends in Fig. 4 were changed accordingly. As suggested, we've revised the conclusions to better discuss a link of the industrial sources identified from a factor analysis based on atmospheric observations to the SPARC bottom-up inventory-based estimations. The revised conclusions now read: "A factor analysis combining the observed concentration enhancements of 18 species was used to identify key industrial sources for CCl4 emissions and to link our atmospheric observation based top-down identification of potential sources with bottom-up inventory-based estimates (e.g., Liang et al.,

2016; Sherry et al., 2017). Three major source categories accounting for 89±6 % of CCl4 enhancements observed at GSN were identified as advertent or inadvertent co-production and escape of CCl4 from CH3Cl production plants (factor (A)) and during industrial C2Cl4 production (factor (C)), and fugitive emissions (factor (B)) from feed-stock use for the production of other chlorinated compounds (e.g., CHCl3) and process agent use. These sources are largely consistent with the bottom-up CCl4 emissions pathways identified in SPARC (Liang et al., 2016). The SPARC estimate of global CCl4 emissions from chloromethanes and PCE plants (pathway B from Liang et al. (2016) and Sherry et al. (2018)) was 13 Gg yr-1, as the most significant source. Fugitive feedstock/process agent emissions, denoted as pathway A by Liang et al. (2016) and Sherry et al. (2018), were estimated to be ∼2 Gg yr-1. These emissions for path-ways B and A had contributions from China of 6.6 Gg yr-1 and 0.7 Gg yr-1, respec-tively. If we assume that the emission rates from the sources correspond to the relative contributions of the corresponding source factors to the total Chinese emission rate (23.6±7.1 Gg yr-1 for the years 2011–2015), source factors (A), CCl4 emissions from chloromethane plants, and (C), emissions from PCE plants, amount to 13±4 Gg yr-1 for China. This is as high as the global bottom-up number of 13 Gg yr-1 for pathway B emissions, and more than 50% higher than Chinese estimate of 6.6 Gg yr-1. This could point to a higher than assumed ratio of CCl4 being emitted from these processes into the atmosphere, although factor (C) could possibly include influence of fugitive emis-sions as a chlorination feedstock for PCE production. Furthermore, also source factor (B), fugitive feedstock emissions are estimated at ∼7±2 Gg yr-1 from China alone, which again contrasts with Chinese estimate of ∼0.7 Gg yr-1 and even a lower global estimate of only 2 Gg yr-1 for pathway A from Liang et al. (2016) and Sherry et al. (2018)."

The paper is fairly well written, but many of the current paragraphs need to be broken up into more distinct sections or primary thoughts. The extended paragraphs of the current version obscure the thoughts, logic of the paper, and the overall content of the text.

»> Based on the reviewer's perspective, we realized that discussions should have been better structured in various places in the previous version breaking a long body of paragraph by a single topic. We do think the revised manuscript has been improved according to reviewer's suggestions. Thanks for the reviewer's editorial comments!

For example, the 2nd para of the Introduction (P2, 4-30) talks about top down emissions, bottom up emissions....I would break this up into paras on: 1) top down emissions (4-12); 2) a SPARC bottom up para (12-16); and 3) a discussion of regional emissions.

»> As the reviewer suggested, we divided this long paragraph into three to make it easier to follow. First, we started with a discussion about the updated bottom-up emissions in the SPARC report, and introduced the global top-down and hemispheric gradient top-down emissions, pointing out that the revised bottom-up estimate of 25 Gg yr-1 is still lower than the average SPARC-merged top-down emission estimate of 35±16 Gg yr-1. Then we added the summed regional emissions estimate from Australia, East Asia, U.S. and Western Europe, and mentioned its lowering than the global total and the relative significance of East Asia contribution.

In the 1st para of section 3 (P. 4 line 18 to P.5 line 32 - 46 lines!), there are a broad range of paragraph thoughts. The paragraph starts with a discuss of the interspecies correlation and ends with a thought on an underestimate of Chinese emissions. Please break this up to improve the flow of the text.

»> We have broken up the original, long paragraph, which is now in section 4 of the revised version, into four paragraphs corresponding to "introduction to an interspecies correlation method", "a reference compound and its emission estimate", "determination of the empirical correlations between the observed enhancements of CCl4 and reference, HCFC-22" and "comparison of the annual CCl4 emissions in China estimated in present study with previous results".

The "Data Overview" section both discusses the data and shows results. I would restructure sections 2 and 3 into: a data, methods, and results sections. The Supporting Information ought to flow better into these data and methods sections.

»> We have completely restructured section 2 of the manuscript by breaking it up into two sub-sections (2.1. Measurements of CCl4 at Gosan and 2.2. Results), and one independent section (3. Potential source regions of CCl4 in East Asia). The new section 3 is comprised of three paragraphs: introduction to trajectory statistics as a tool to illustrate the regional distribution of potential CCl4; input data and conditions for calculation; and description of the resulting map of potential source areas. We've also added specific information on corresponding SI text accordingly in the new section 3. Air mass source country classification that had been discussed in the last paragraph of Data overview section in the original manuscript, now moved to the beginning of section 4, as a transitional paragraph to the following country-specific emission analysis.

Again, break up the single paragraph of the conclusions into short paragraphs. The main messages are lost in this "run-on" paragraph.

»> We have re-organized the conclusions with four short paragraphs. We hope this can convey ideas more clearly to readers. For the text revision, please see the earlier response.

Figures are good. For Fig. 4, put some vertical lines on the plot to see how the bars line up with chemical names at the bottom.

»> Done

Fig. S5. What are the colors for? Do they indicate statistical significance?

»> We now say in the figure caption: "The colors by shade indicate statistical significance."

Please also note the supplement to this comment:
https://www.atmos-chem-phys-discuss.net/acp-2018-220/acp-2018-220-AC1-

supplement.pdf

---

## Author Comment (AC2) · 19 Jun 2018

Referees' comments on "Toward resolving the budget discrepancy of ozone-depleting CCl4: An analysis of top-down emissions from China" by Sunyoung Park, Shanlan Li, Jens Mühle, Simon O'Doherty, Ray F. Weiss, Xuekun Fang, Stefan Reimann, Ronald G. Prinn

We thank the referees for their thoughtful and thorough reviews. We are pleased that all the reviewers see our manuscript as a valuable contribution to the field. We have made changes to the manuscript to answer the suggestions of the reviewers and clarified a few points raised in review. We respond to the referee's comments below and a revised

[Figure]

version of the manuscript including most of the changes suggested by the reviewers will be submitted to the editor. We thank the reviewers and the editor for their time and effort and appreciate the recommendation for publication in Atmospheric Physics and Chemistry.

Reviewer comments:

Referee #2:

Park et al presented a top-down emissions estimate of CCl4 from East Asia based on high frequency surface measurements of halocarbons at the Gosan sites. This paper is timely. Results presented in this paper provide crucial pieces of information that closes the CCl4 global budget as well as providing the atmospheric observational evidence that unreported CCl4 emissions during chloromethans and PCE production. However, the writing in many places can use some improvements. I recommend the authors go through the entire manuscript thoroughly to improve the clarity and accuracy. The paper should be published in ACP after the following comments are addressed.

1. P1 L15, "the 2010" -> "a 2010"

»> Done

2. P2 L5-7. You should state that the global top-down emissions are derived based on both the CCl4 lifetimes and the observed global decline rate.

»> A point well-taken. We have changed it to what the reviewer suggested (underlined words are the edits): "To verify these reported bottom-up estimates, independent CCl4 emission studies have used the total lifetime of CCl4, atmospheric observations, i.e., the observed decline rate of CCl4 concentrations and atmospheric transport models to derive "top-down" estimates."

3. P2 L9. The global emissions number from Liang et al, 2014 was 39Gg/yr, not 30Gg/yr.

[Figure]

»> We realized from the reviewer's comments that the citation was incorrect. The 39 Gg/yr emission from Liang et al. (2014) had been updated into the value of 30 Gg/yr with the new 33-year lifetime of CCl4 in the SPARC report (Liang et al., 2016). So we've changed the original sentence into the following to clarify the updated estimate: "A recent top-down study based upon the observed temporal trend and inter-hemispheric gradient of atmospheric CCl4 (Liang et al., 2014) consistently derived global CCl4 emissions of $30\pm5$ Gg yr-1 in 2000–2012 with the newly determined relative strength of the oceanic sink vs. the soil loss (Liang et al., 2016)."

4. P2 L11-12. I am not sure why you say "unidentified sources and/or unreported anthropogenic emissions". CCl4 is a predominantly man-made compound, therefore the emissions sources are anthropogenic.

»> Agreed. For clarification the word "anthropogenic" has been edited into "industrial". We think unidentified old, contaminated soils and/or facilities can be "unidentified sources" here.

5. In many places, need to change the "," after the references to ";".

»> Corrected

6. P2 L27-30. You need to merge these two sentences and present the results from these studies in a less confusing way with a correct referencing style. In the present form, it is hard for the readers to figure out from which studies the 4.3 and 5.2 Gg/yr were from.

»> The sentences have been merged and edited to clarify that those numbers were updated as new bottom-up emission estimates (underlined words are the edits): "Most recently, Bie et al. (2017) published post-2010 bottom-up emission estimates for China of 4.3 (1.9–8.0) Gg yr-1 in 2011 and 5.2 (2.4–8.8) Gg yr-1 in 2014, which updated a previous estimate of zero emission (Wan et al., 2009) by including the conversion of C2Cl4 emissions to CCl4 as well as a source of CCl4 from coal combustion smog."

7. P2 L30. Change to "8-year continuous high frequency, high precision atmospheric CCl4 concentrations measured…."

»> Changed

8. P3 L2. Change "below the " to "to the south of .."

»> Changed

9. P3 L7. I am not sure what do you mean by "well situated to allow monitoring of long-range transport from the surrounding region". Is this because of elevation or it is in remote clean ocean? By surrounding region, what regions are you referring to? China? The Korean Peninsula? Please clarify.

»> We now provide more explicit description of the station in the Supplementary Information as well as give more information in the figure caption (Fig. S1): "Gosan station (GSN, 33.25°N, 126.19°E, Jeju Island, Korea) located on the boundary between the Pacific Ocean and the Asian continent (Fig. S1) is characterized by warm wet East Asian Summer Monsoon and cold dry winter, and by distinct seasonal wind patterns with strong northern winds in winter, and southern influence during summer. These wind patterns are favorable for monitoring air masses passing through East Asia, especially China and Korea. Clean background conditions are observed when a clean stream of air flows in directly from northern Siberia in winter and during transport of southerly oceanic winds in summer (Fig. S2)."; "Fig. S1. The Gosan AGAGE (Advanced Global Atmospheric Gases Experiment) station is located atop a 72-m cliff (air intake elevation: 89 meter above sea level) on the remote south-western tip of Jeju Island, 100 km south of the Korean peninsula, allowing for monitoring of long-range transport from the surrounding region."

10. P3 L10. Please include the actual values than just say "high-precision and high frequency"

»> The data frequency has been given as "every two hours from 2008 to 2015" and the

experimental precision has also been stated in the sentence: "Precisions ($1\sigma$) derived from repeated analysis (n = 12) of a working standard of ambient air are better than 1 % of background atmospheric concentrations for all the compounds, e.g. $\pm$ 0.8 ppt ($1\sigma$) for 85.2 ppt of CCl4."

11. P3 L22. You need to define what do you mean by "baseline values". This is jargon.

»> We have added the following text in parentheses: "(i.e., background values representing regional clean condition without regional/local pollution events)".

12. P4 L6-7. It would be good to add references here.

»> We've added the website http://eng.chinaiol.com/, where the locations of the main factories producing HFCs, HCFC-22 and fluorocarbons are given. The locations were also denoted in Fig. S9.

13. P7 L11. It is interesting that CFC-11 showed up in the source factor. Does this indicate that CFC-11 is also produced in the CM plants?

»> Given the fact that CFC-11 can be readily produced by the reaction of by-produced impurity, CCl4 with HF, the observed high contribution of CFC-11 in the fugitive emissions group is explainable in association with production of chloromethanes and their feedstock use for fluorinated compounds. For further comments regarding recent enhancements of CFC-11 observed at Gosan, please see the last response below.

14. P7. It will be of great value to CCl4 source identification to link the discussions in the source factors to the industrial production, usage, and potential emissions pathway in Sherry et al. (2017). Such a discussion will help to build link from bottom-up inventory-based estimate to atmospheric observation based top-down estimate.

»> Agreed. According to the reviewer's suggestion, we've revised the conclusions to better discuss a link of the industrial sources identified from a factor analysis based on atmospheric observations to the SPARC bottom-up inventory-based estimations. The revised conclusions now read: "A factor analysis combining the observed concentration enhancements of 18 species was used to identify key industrial sources for CCl4 emissions and to link our atmospheric observation based top-down identification of potential sources with bottom-up inventory-based estimates (e.g., Liang et al., 2016; Sherry et al., 2017). Three major source categories accounting for 89±6 % of CCl4 enhancements observed at GSN were identified as advertent or inadvertent co-production and escape of CCl4 from CH3Cl production plants (factor (A)) and during industrial C2Cl4 production (factor (C)), and fugitive emissions (factor (B)) from feedstock use for the production of other chlorinated compounds (e.g., CHCl3) and process agent use. These sources are largely consistent with the bottom-up CCl4 emissions pathways identified in SPARC (Liang et al., 2016). The SPARC estimate of global CCl4 emissions from chloromethanes and PCE plants (pathway B from Liang et al. (2016) and Sherry et al. (2018)) was 13 Gg yr-1, as the most significant source. Fugitive feedstock/process agent emissions, denoted as pathway A by Liang et al. (2016) and Sherry et al. (2018), were estimated to be ∼2 Gg yr-1. These emissions for pathways B and A had contributions from China of 6.6 Gg yr-1 and 0.7 Gg yr-1, respectively. If we assume that the emission rates from the sources correspond to the relative contributions of the corresponding source factors to the total Chinese emission rate (23.6±7.1 Gg yr-1 for the years 2011–2015), source factors (A), CCl4 emissions from chloromethane plants, and (C), emissions from PCE plants, amount to 13±4 Gg yr-1 for China. This is as high as the global bottom-up number of 13 Gg yr-1 for pathway B emissions, and more than 50% higher than Chinese estimate of 6.6 Gg yr-1. This could point to a higher than assumed ratio of CCl4 being emitted from these processes into the atmosphere, although factor (C) could possibly include influence of fugitive emissions as a chlorination feedstock for PCE production. Furthermore, also source factor (B), fugitive feedstock emissions are estimated at ∼7±2 Gg yr-1 from China alone, which again contrasts with Chinese estimate of ∼0.7 Gg yr-1 and even a lower global estimate of only 2 Gg yr-1 for pathway A from Liang et al. (2016) and Sherry et al. (2018)."

15. Figure 3 and related discussions. (1) I wonder if part of the difference between the

Vollmer et al., 2009 and this study is due to the location of Gosan vs. Shandianzi. The location of Gosan captures most of the outflow from the industrial central and south China, where all the CCl4 production industries are located (as suggested by Figure 2), while Shandianzi captures mostly the air influenced by N. China, without much CM production. Should consider add a related discussion on this in the manuscript.

»> Yes, this is an important point to mention. We agree with the reviewer that difference in the location of monitoring sites and thus in their footprint distributions of compounds of interest must be one of potential reasons for discrepancies found between emission estimates derived from different monitoring sites. Interestingly, however, the CCl4 emission rate of 16.8 ± 5.6 Gg yr-1 in 2008 we derived in this study was statistically consistent with the 2007 emission rate of 15 (10–22) Gg yr-1 given in Vollmer et al. (2009) within their uncertainties. The agreement could be coincidental, but it could also be consistent with the fact that even though the CM-related production facilities are more likely located in industrial central and south China (Fig 2 and Fig S9), the increase in both the feedstock production sector of CCl4 and emissions from CCl4 by-production was reported only since 2011, i.e. post-2010 (Bie et al., 2017: see Fig. 2 in the paper). In this respect, it is possible that the 2007 emission estimate derived from Shandianzi and the 2008 estimate from and Gosan were not much different, even if Shandianzi is known to capture mostly the air masses influenced by north China – covering most down to Shandong and Anhui for CCl4 (Vollmer et al., 2009) and to Jiangsu and Anhui for CO (An et al., 2014), and thus could possibly miss the influences from Henan, Hubei, and Guangdong provinces. Therefore, it seems that further discussion about potential differences in emissions estimate for CCl4 between Gosan vs. Shandianzi, particularly in relation to the location of CCl4 emission sources can be made when further analysis on the CCl4 data and results of post-2010 from Shandianzi are published.

Reference: An, X., Yao, B., Li, Y., Li, N., Lingxi Zhou, L.: Tracking source area of Shangdianzi station using Lagrangian particle dispersion model of FLEXPART, Meteorol. Appl. 21: 466–473, 2014.

(2) The covariance of CFC-11 and CMs (source factor 2) is very interesting. Does this mean CFC-11 is also an intendended by-product during the industrial process and the recent increase in CFC-11 unreported emissions (Montzka et al. 2018) is to some extent linked to the CCl4 emissions increase in China between 2012-2016?

»> As we noted in response to the comment above regarding the high contribution of CFC-11 shown in the fugitive emissions group, CFC-11 can be readily produced by the reaction of by-produced impurity, CCl4 with HF and thus it would be possible that the observed high contribution of CFC-11 in the fugitive emissions group could be association with production of chloromethanes and their feedstock use for fluorinated compounds, whether it is intended or not Recent increase in unreported CFC-11 emissions discussed in Montzka et al. (2018) is indeed consistent with recent enhancements in CFC-11 pollution signals observed at Gosan (see figures below). It would also be possible that these enhancements might be associated with production of many fluorinated compounds using chloromethanes as feedstock and thus with persistent CCl4 emissions in East Asia, as shown in this study. This reviewer's question is one of the most important issues these days. So, if allowed we'd like to complete a separate analysis for CFC-11enhancements at Gosan and address this issue further in another manuscript.

Please also note the supplement to this comment:
https://www.atmos-chem-phys-discuss.net/acp-2018-220/acp-2018-220-AC2-supplement.pdf

[Figure]

**Fig. 1.**

[Figure]

**Fig. 2.**

---

## Author Comment (AC3) · 19 Jun 2018

Referees' comments on "Toward resolving the budget discrepancy of ozone-depleting CCl4: An analysis of top-down emissions from China" by Sunyoung Park, Shanlan Li, Jens Mühle, Simon O'Doherty, Ray F. Weiss, Xuekun Fang, Stefan Reimann, Ronald G. Prinn

We thank the referees for their thoughtful and thorough reviews. We are pleased that all the reviewers see our manuscript as a valuable contribution to the field. We have made changes to the manuscript to answer the suggestions of the reviewers and clarified a few points raised in review. We respond to the referee's comments below and a revised

[Figure]

version of the manuscript including most of the changes suggested by the reviewers will be submitted to the editor. We thank the reviewers and the editor for their time and effort and appreciate the recommendation for publication in Atmospheric Physics and Chemistry.

Reviewer comments:

Referee #3: There has been a long-standing mystery of why the atmospheric concentration of carbon tetrachloride has declined much slower than predicted after its use was banned by the Montreal Protocol. The SPARC (2016) report resolved only part of this mystery by assessing a slightly longer atmospheric lifetime and by increasing estimates of industrial bottom-up emissions. However, a reconciliation of the top-down and bottom-up estimates was not achievable unless the error bars were stretched to their limits.

The present study by Park et al. utilizes high precision measurements of a suite of halocarbons at a background air monitoring station at Gosan, South Korea, to identify the origins of large fugitive emissions of CCl4 and to estimate their overall emission rates between 2008-2015. The analysis determines that emissions from heavily industrialized regions of China can account for roughly 24 +/- 7 Gg/yr CCl4 between 2011 and 2015 instead of the 4-5 Gg/yr reported bottom-up emissions rates. Surprisingly, emission rates do not seem to have declined over this time period. The additional 19 Gg/yr of fugitive emissions from China would account for over half of the global CCl4 emissions, and perhaps be enough to resolve the remaining mystery of carbon tetrachloride. Thus, this paper represents a very important scientific advance indeed.

The atmospheric measurements are of high quality and the method of using back air trajectories combined with empirical correlations with a reference compound (HCFC-22) is supported by an independent derivation of HCFC-22 emissions that agrees with prior estimates. The industrial source apportionment using the Positive Matrix Factorization (PMF) model yielded several strong relationships, pointing to multiple sources

of CCl4 associated largely with emissions with other compounds. The interpretation is that the fugitive emissions are occurring at the factory level during production of various chlorocarbons. This seems highly plausible, as the production of these compounds are co-located, whereas the consumption of these compounds are expected to be more widely distributed.

Overall, the writing and figures are clear, and the methodology maximizes the functionality of a high quality dataset. I encourage the publication of this important work, with only a few minor edits suggested below.

1. Pg 2, line 6. The relevant soil sink reference is: Rhew & Happell, 2016, not Rhew et al., 2008.

»> Changed. Thanks much!

2. Pg 3, line 10. Here it would be helpful to have a reference or more description about the Gosan station. A brief description of the sample intake line, its height and its proximity to other major landscape features would be helpful details.

»> We now provide more explicit description of the station in the Supplementary Information as well as give more information in the figure caption (Fig. S1): "Gosan station (GSN, 33.25°N, 126.19°E, Jeju Island, Korea) located on the boundary between the Pacific Ocean and the Asian continent (Fig. S1) is characterized by warm wet East Asian Summer Monsoon and cold dry winter, and by distinct seasonal wind patterns with strong northern winds in winter, and southern influence during summer. These wind patterns are favorable for monitoring air masses passing through East Asia, especially China and Korea. Clean background conditions are observed when a clean stream of air flows in directly from northern Siberia in winter and during transport of southerly oceanic winds in summer (Fig. S2)."; "Fig. S1. The Gosan AGAGE (Advanced Global Atmospheric Gases Experiment) station is located atop a 72-m cliff (air intake elevation: 89 meter above sea level) on the remote south-western tip of Jeju Island, 100 km south of the Korean peninsula, allowing for monitoring of long-range

transport from the surrounding region."

3. Pg 3, line 20. The authors should specify that the remote background station in the Northern Hemisphere is Mace Head, Ireland.

»> We've specified Mace Head station as a NH remote monitoring site (underlined words are the edits): "Note that the "background" concentrations at GSN agree well with the background concentrations observed at the Mace Head station (53°N, 10°W) in Ireland that represents a remote background monitoring station in the Northern Hemisphere and are declining at a similar rate to the global trend (Fig. S4)."

On a related note, it appears that no other AGAGE station comes anywhere close to the pollution level events that Gosan station experiences. Expressing this, perhaps in a quantitative way (standard deviation?) would add to the argument that the Gosan station is uniquely situated among the network to capture the primary region of fugitive emissions. After seeing the data published online from all the other stations, it seems clear that this is so.

»> A point well-taken. We've revised the description about the time series plot of the atmospheric $CCl_4$ concentrations observed at Gosan (Fig. 1) in the section 2 (underlined words are the edits): "The 8-year observational record of $CCl_4$ analyzed in this study is shown in Fig. 1. It is apparent that pollution events (red dots) with significant enhancements above "background" levels (black dots) occur frequently, resulting in daily variations of observed concentrations with relative standard deviations (RSDs) of 4–20% in contrast to the RSDs of 0.1–1.5% shown in all the remote stations operated under the AGAGE program. It clearly implies ongoing emission of $CCl_4$ in East Asia."

4. Section 4. Although the time periods may differ, it may be useful to compare these results with some ground based measurements within China that are closer to the source regions. For example, prior studies have found very high concentrations of halocarbons in the Pearl River Delta region of China. Zhang et al., (JGR 115, D15309,2010)

measured elevated concentrations in 2007 and report "The high correlation between CCl4 and CFCs suggests that this source was more related to the production than the consumption of refrigerants." How important is the Pearl River Delta region compared to other regions in the present study? It is difficult to assess based on the maps.

»> This is a good suggestion. The Pearl River Delta (PRD) region denoted by blue circles in Fig. S9 shown below is one of important source regions in China.

Referring to the reviewer's comments on Zhang et al. (2010), we've added the following sentences after the discussion about the potential source distributions (Fig. S9) in the section 5 (5. Industrial source apportionment of atmospheric CCl4 in East Asia): "Our results are also consistent with a previous study on halocarbons observations in the Pearl River Delta region of Guangdong (Zhang et al., 2010), which revealed using a source profile analysis that the emissions of CFCs and CCl4 from an industrial source related to chemical (i.e., refrigerant) production increased by 1.4–2.0 times from 2001–2002 to 2007, even though atmospheric mixing ratios of these compounds did not change much for the 6 years. It implied increased use of CCl4 in chemical productions."

5. Section 4. It would be interesting to see if CH3Br adds any clarity to the model – it is not shown in Figure 4 but shows a high correlation to many other compounds (Figure S5), including CCl4. CH3Br is also banned by the Montreal Protocol but has substantial natural as well as anthropogenic sources. As there are no major natural sources of CCl4, the elevated concentrations of CH3Br may be associated with previously unknown anthropogenic sources. It may be outside the scope of this particular paper, but it would be worth investigating if CH3Br is also emitted from CH3Cl production sites.

»> A very interesting suggestion. The time series of atmospheric CH3Br concentrations in 2008–2015 at Gosan shows below the continuous concentration enhancements as high as ∼30 ppt. As the reviewer mentioned, the observed enhancements of CH3Br are also in a high correlation to many other anthropogenic compounds (now shown in Fig. S6) but are in a poor correlation with CHBr3 (not shown) - an ocean

source tracer. This suggested negligible influence of oceanic source but consistent emissions from nearby fumigant-related source regions.

One previous study in my group estimated CH3Br emission from East Asia to be 6.5 (4.8–8.9) Gg yr-1 based on atmospheric CH3Br concentrations observed from Nov. 2007 to Dec. 2008 at Gosan (Li et al., 2011). This contributed to 50% of the global emission for 1996–2007 (13.8 Gg yr-1, Yvon-Lewis et al., 2009) from "fumigation-quarantine and pre-shipment" derived based on government and industry statistics (UNEP Methyl Bromide Technical Options Committee, 2006). Later, in a following study (Li et al., 2014) we applied a positive matrix factorization (PMF) model to the enhanced concentrations of 18 halogenated compounds including CH3Br obtained for the period from Nov. 2007 to Dec. 2011 and found that CH3Br was grouped in a separate factor from other compounds (see the left panel below). In addition, potential source region analysis revealed that the factor distinguished by a high contribution of CH3Br was predominant along the coastal area in Korea, Yangtze river delta region, and near the Vladivostok. Therefore, the high contribution of CH3Br in the factor was most likely explained by fumigation use in "quarantine" and "pre-shipment" treatments (QPS), which is exempt for all countries under the Montreal Protocol. Since we could not notice any change in the observed enhancements of CH3Br when comparing Nov. 2007–2011 vs. 2012–2015 data and thus expected a separate factor for QPS identified by dominant contribution of CH3Br in PMF results, we excluded CH3Br in the PMF analysis to simplify the results and to better focus on CCl4 related factors. Nonetheless, as the reviewer suggested, it must be worth monitoring if CH3Br could be categorized together with industrially-emitted chemical compounds in future.

Reference: Yvon-Lewis, S.A., Saltzman, E. S., Montzka, S. A.: Recent trends in atmospheric methyl bromide analysis of post-Montreal Protocol variability, Atmos. Chem. Phys., 9, 5963–5974, 2009.

6. Pg 9, line 4. The data repository for the Gosan dataset will need to be updated, as the website specified does not appear to have accessible data repositories.

»> We've updated the data repository by specifying the sub-folder on the website, http://agage.eas.gatech.edu/data_archive/agage/gc-ms-medusa/

7. Figure S1. The Gosan station should be highlighted with a larger symbol. Also: the dark blue obscures the text and border slightly.

»> The station location was emphasized with a star symbol. The border line and texts were moved in front of the plots to minimize their obscury. While updating, we realized that the analysis period stated in the original figure caption was wrong, and it was corrected: "2008" to "2008-2015".

8. Figure S5: The color scheme helps, but the text is very hard to read. Please make the graphic large enough such that the numbers are readable. It appears that the image can potentially be increased 25% in size while still fitting in the margins. Subscripts can also be added to the left side labels.

»> The figure was updated by enlarging the image along with a bigger font size for numbers. The Y labels were also corrected with subscripts

9. Figure S7. Why is 2010 in bold and red? »> In developing countries, the regulations on production and consumption of $CCl_4$ started to go into effect in 2010. We'd intended to indicate the phase-out year in yearly correlation slopes. The following sentence has been added in the figure caption: "Note that $CCl_4$ production and consumption for dispersive applications in developing countries were phased out in 2010".

10. Figure S8. What do the colors of the legend indicate?

»> The unit of ppt was added in the color scale.

Please also note the supplement to this comment:
https://www.atmos-chem-phys-discuss.net/acp-2018-220/acp-2018-220-AC3-
supplement.pdf

[Figure]

[Figure]

**Fig. 1.**

[Figure]

Fig. 2.

[Figure]

**Fig. 3.**

[Figure]

**Fig. 4.**

---

## Author Comment (AC4) · 19 Jun 2018

Referees' comments on "Toward resolving the budget discrepancy of ozone-depleting CCl4: An analysis of top-down emissions from China" by Sunyoung Park, Shanlan Li, Jens Mühle, Simon O'Doherty, Ray F. Weiss, Xuekun Fang, Stefan Reimann, Ronald G. Prinn

Interactive comments:

A. McCulloch archie.mcculloch@btinternet.com

In many respects this is an excellent paper that uses high frequency atmospheric measurements that have low and well defined uncertainty to a) estimate the magnitude of carbon tetrachloride emissions in air reaching the Gosan observatory and b) characterize the geographical location of those emissions. However, the authors go several steps too far when they assign physical processes to the sources of releases of CCl4.

»> We thank Dr. McCulloch for his thoughtful and thorough comments. We are very pleased that he considered our manuscript as a valuable contribution to the field. Based on Dr. McCulloch's perspective, we realize possible confusion over terminology (e.g., fugitive emissions from feedstock use vs. escape of by-product), which might mislead the readers. We are very careful with our definitions and terminology throughout the manuscript. We have also made changes to the manuscript to answer the suggestions of the reviewers. We hope that we've clarified a few points and concerns raised in Dr. McCulloch's comments below and have satisfactorily addressed these concerns in the revised version of the manuscript.

Showing where the CCl4 is emitted in China is a major contribution to our understanding of the input of CCl4 to the atmosphere but to then assign emissions to particular industrial processes based solely on correlation with emissions of other chemicals is an over-interpretation of the data presented and contradicts practical evidence used to build bottom-up inventories for both China and the world as a whole. Consequently, the discussion of results and conclusions on pages 7 to 9 needs to be completely rewritten taking into account that the measurements relate to emissions and do not relate to contemporaneous production in either quantity or location.

»> It appears that there is a misunderstanding in that "to assign emissions to particular industrial processes based solely on correlation with emissions of other chemicals is an over-interpretation of the data presented." We did use the correlation with HCFC-22 exclusively for estimating the total Chinese emissions of CCl4, based on HCFC-22 emission numbers for China. For the assignment of the CCl4 emissions to particular industrial processes (e.g., pathways A vs. B) the results from a far more sophisticated PMF receptor model was used. The PMF was applied to the enhanced concentrations

(i.e, in which the influence of the regional background levels on the measurements has been removed by subtracting baseline values) of all 18 species for their pollution source apportionment. Each resulting source category (i.e., factor) was characterized by a distinctive combination of high-contribution species and was defined accordingly based on the source profiles. For the phrase of "to assign emissions to particular industrial processes …. contradicts practical evidence used to build bottom-up inventories for both China and the world as a whole.", our analysis identified three major source categories of (A) CH3Cl production plants (B) fugitive emissions from feedstock use for the production of other chlorinated compounds (e.g., CHCl3) and from process agent use, and (C) PCE production plants. These sources are largely consistent with the bottom-up CCl4 emissions pathways identified in SPARC (Liang et al., 2016; Sherry et al., 2018). For the sentence of "the measurements relate to emissions and do not relate to contemporaneous production in either quantity or location", we agree that the observed concentrations are related to emission and not directly to contemporaneous production. It was the reason why we used a factor analysis on the measurements to suggest source apportionment for industry-related compounds. The PMF results pointed to a few source categories of CCl4, which were defined based on the source profiles associated with other compounds. Since we note the production and feedstock use of those compounds can be co-located, whereas general consumption and legacy release are more widely distributed, we believe our interpretation CCl4 emissions are occurring at the factory level during production of various halocarbons is highly plausible. In addition, one of primary motivations for this study is to suggest first observational based estimates for industrial sources in comparison with the SPARC industrial bottom-up emissions, thereby providing insight into a link between the different approaches. Thus, the discussion of results and conclusions on pages 7 to 9 in the original text served the distinct and deliberate purpose and thus we have left the main messages in the discussion as they were.

The SPARC report on CCl4 [SPARC, 2016] and subsequently Sherry et al. [2018] showed the principal sources of emission to be inadvertent and unreported, arising

from: manufacture and use of chlorine, including as a disinfectant; leakage from historic landfill; unreported emissions of material produced during the manufacture of legitimate chemical products and then used in emissive applications in contravention of regulations. Of the 25 Gg/y emissions estimated [SPARC, 2016] only 2 Gg/y were calculated to be fugitive emissions. Actual fugitive emissions, that is emissions from the plant during the production of CCl4 (either as desired product or as by-product) or its use as a chemical feedstock, are better understood than any of the other sources and account for only a few percent of the global emissions of CCl4.

»> For our current understanding of the fugitive emissions, we found a bit difference between the SPARC report (and Sherry et al.) vs. your statement here of "Actual fugitive emissions . . . are better understood than any of the other sources and account for only a few percent of the global emissions of CCl4." Fugitive feedstock/process agents emissions (SPARC pathway A) derived based upon UNEP reported, feedstock emission factors of 0.5–2% and assumed 95% fraction of total production for feedstock use according to the post-1995 reporting to UNEP (Montzka and Reimann, 2011), were estimated to be ∼2 Gg yr-1. The underlying assumption is that (1) UNEP reported total production and the reported amounts used as feedstocks have completed globally and regionally even after recent rapid increase in production of various halocarbon compounds and (2) the approximation of 0.5% to 2% leakage from contained feedstock production is accurate. We also note that the fugitive emissions defined in both the SPARC report (pathway A) and Sherry et al. (pathway C) include not only feedstock usage, but also process agents use.

There is an important distinction between fugitive emissions, which are (or at least can be) controlled and which are an economic loss to the manufacturer and material that is disposed of "usefully" but which is, in the end, emitted into the atmosphere.

»> For clarification we described our factor (A), which was now re-named to make it easier to compare them to the SPARC report as asked for by one of reviewers, as advertent or inadvertent co-production and escape of CCl4 during CH3Cl generation

in the chemical plants.

At the root of the problem with the paper is the naive assertion that all emissions arise from chemical production. Among the 26 halocarbons reported in the paper, this is true only for: HFC-23 (which arises from HCFC-22 production and where 90% of emissions are from the chemical plant itself [Simmonds et al., 2018]) and that part of CF4 emissions that arise in aluminium production.

»> We did not intend to give a wrong impression that "all emissions of 26 halocarbons arise from chemical production". Addressing emission sources of individual "26" species is certainly beyond the scope of this study. As discussed in the section of "Industrial source apportionment of atmospheric CCl4 in East Asia", only CCl4, CH3Cl, CH2Cl2, CHCl3, CFC-11, HFC-23 and CH2Cl2 of 18 species analyzed in the PMF model were discussed in terms of association with chemical manufacturing of various halogenated compounds, because importantly, it turned out that these compounds were grouped together in a couple of source factors.

For the rest, emissions occur during use of the substance or when equipment containing the substance is scrapped at the end of its life. This is well known, indeed it forms the basis for the national estimates of fluorinated greenhouse gases submitted to UNFCCC, and the authors have no grounds for asserting otherwise.

»> As shown in the section of "Industrial source apportionment of atmospheric CCl4 in East Asia", potential emissions occurring during use, recycling and old equipment destruction were discussed in separate source factors: "refrigerant use" and "foam blowing", in which we noticed no CCl4 contribution (see SI text).

From the point of view of assigning geographical locations of CCl4 emissions, the important species are methyl chloride, dichloromethane, tetrachloroethene and HFC-23. These form part of the suite of chemicals used by the authors in the analysis of the observation to show how emissions of CCl4 are geographically co-located with enhanced emissions of other species, with the results shown in Figure 4.

»> Yes, it is an important point that the production and feedstock use of the suite of chemicals can be co-located, whereas general consumption and legacy release are more widely distributed. Therefore, the interpretation of the results shown in Fig. 4 that the emissions of CCl4 as co-product/by-product and as feedstock/process agent occur at production level is reasonable.

Combustion processes in which the fuel contains chloride ion (much of Chinese coal, municipal solid waste and biomass) account for most of the methyl chloride emitted into the atmosphere by human activity and there is a substantial natural source (five times larger than anthropogenic) [Carpenter & Reimann, 2014]. Sherry et al. [2018] show 26 Gg/y of methyl chloride is made in China from CCl4, so the statement in Figure 4, panel 1 that 38% of release is due to methyl chloride production implies a fugitive emission rate of 35% - ludicrous both from economic and public health points of view.

»> It appears that the numbers above need to be clarified. 38% of the CCl4 enhancements from source factor (A) and 19% of CCl4 from factor (C) correspond to advertent or inadvertent co-production and escape of CCl4 during CH3Cl and C2Cl4 production in the chemical plants. They are described as CM plants and PCE/CTC plants in the section of CTC production in Sherry et al. (2018), respectively. Therefore, as stated in the conclusion, if assuming that those relative contributions of the factors correspond to the source emissions, the unreported, non-feedstock emissions from CCl4 production amount to ∼13 Gg yr-1 of CCl4 in China alone. Then the emission rate of CCl4 would be ∼6% of 203 Gg yr-1 for global CCl4 production, and ∼17% of 77 Gg yr-1 for CCl4 production in China (Sherry et al., 2018), with the assumption that the UNEP reported amounts are complete.

In Figure 4, panel 2, 32% of CCl4 emissions are co-located with 78% of DCM emissions. DCM has a high solvent power for oils and greases and for some polymeric materials. These properties, coupled with its volatility (boiling point 40.1oC) have led to its wide use as an industrial solvent in applications such chemicals and pharmaceuticals production and to a lower extent as food extraction solvent, for metal cleaning

and paint removal. It is also a component of special adhesives and has been used in PU foam blowing, in aerosols, paint strippers and as laboratory agent. Many of these uses can result in much of the DCM employed being emitted into the environment (so-called emissive uses). More recently, smaller quantities of DCM have been used as chemical feedstock to produce HFC-32 ($CH_2F_2$, difluoromethane) but use as feedstock does not result in significant emission of DCM. The conclusion must be that 32% of the CCl4 emissions found are co-located with the areas where DCM is emitted from its use, principally as a solvent.

»> Yes, our PMF results were also consistent with the fact that CH2Cl2 (DCM) has been widely used as an industrial solvent. The 26$\pm$14 % contribution of CH2Cl2 shown in the factor (G) seems to suggest its use as a cleaning solvent, because the factor was characterized by high contributions of SF6, C2F6, and CF4, and thus explained as the source factor for semiconductor/electronics manufacturing processes (see SI text). Importantly, a large fraction of CH2Cl2 was also shown in the source factor (B) with consistently high contributions of CHCl3 and CCl4, which are known to be produced along with CH2Cl2 as by-products of chlorination, and used as feedstocks or solvents in chemical manufacturing. This factor was also distinguished by a high contribution of HFC-23, a typical by-product of HCFC-22 generation where CHCl3 is used as a feedstock. As CHCl3 and CH2Cl2, short-lived, Cl-contained species are still allowed for emissive uses, their high percentages could be explained in part by ubiquitous industrial use. However, when considering that the observed enhancements of HFC-23, CHCl3, and CH2Cl2 are grouped together into a single factor with CCl4 and CFC-11 that have been banned for emissive (dispersive) use, this factor for CCl4 (also for CFC-11) should represent emissions occurring as desired product/by-product and as feedstock/process agent during chemical manufacturing processes (for the details of CFC-11, see the response to Reviewer #2's comments).

Exactly the same can be said about Figure 4, panel 3. Here the co-location demonstrated is with tetrachloroethene. Like DCM, this is used in many applications as a

solvent, particularly in textile and metal cleaning, which is where most of the emissions arise. Compared to the amount that is produced and used, feedstock emissions are small. So, a further 19% of CCl4 emissions are co-located with solvent emissions of tetrachloroethene.

»> As we stated earlier, PCE (C2Cl2) is also allowed for dispersive uses with various applications as an industrial solvent like CH2Cl2, and thus its emission could be widely distributed geographically. However, the factor (C) was distinguished by both CCl4 and PCE, which means that CCl4 emissions are co-located with PCE emissions, just as commented. In other words, there are possible co-emissions during their productions and consumptions (uses). Then the explanation for CCl4 that is not allowed for its dispersive use should be escape emissions from co-production during industrial C2Cl4 production and in part fugitive emissions of CCl4 being used as a chlorination feedstock for C2Cl4 production, both occurring at the chemical production level. Furthermore, potential source region distributions of the three emission sources for CCl4 shown in Fig. S9 (note its original number was S8) are very similar. It consistently supported the argument that the three sources are more likely related to chemical manufacturing industry including production/by-product production and feedstock/process agent use of related compounds, rather than related to dispersive use and/or legacy emissions, which are expected to be widely distributed with different patterns among the three source factors.

The first two panels of Figure 4 also show that 70% of CCl4 emissions are co-located with 90% of HFC-23 emissions. CCl4 is not directly involved in the chemical processes that make or use HFC-23.

»> HFC-23 is not either produced on purpose or used (only less commonly used in semiconductor fabrication). Therefore, any compounds except HCFC-22 cannot be "directly involved in the chemical processes that make or use HFC-23", because it is a typical by-product of HCFC-22 production. CHCl3 is used as a feedstock for HCFC-22 production. CHCl3 is produced in chloromethanes plants from the reaction of hydrochlorinating methanol (CH3OH) with HCl to form methyl chloride (CH3Cl), and subsequent chlorination to produce CH2Cl2, CHCl3, and CCl4. The reaction of CCl4 with H2 to form CHCl3 and HCl is also possible. In this sense, fugitive emissions of CCl4 as an impure co-product of CHCl3 and/or as feedstock for CHCl3 can be associated with HCFC-22 production, and thus indirectly related to emission of HFC-23.

The authors present no evidence to attribute the co-location of these emissions and those of DCM, chloroform and CFC-11 to "fugitive emissions..............at the factory level during various chemical manufacturing processes in China" (page 7, lines 20-21). This is an assumption that, as discussed above, has no basis in fact.

»> We've interpreted our results, which reveal that the regulated CCl4 has been emitted with significant amounts, 4–5 times larger than the inventory-based bottom-up emissions estimate, and the emitted CCl4 has shared its source factors with CHCl3, CH2Cl2, CH3Cl, HFC-23, CFC-11 and PCE. For a compound like CCl4 fully regulated for dispersive uses, if it is co-emitted with other compounds produced as co-products or by-products of each other and consumed as feedstocks/process agents/solvents in chemical industries, which source could you suggest as an alternative explanation for their co-emissions except emissions occurring at the factory level when CCl4 is produced as a by-product and used as feedstock/process agent during various chemical manufacturing processes?

Figure S5 shows correlations between observations of the atmospheric concentrations of 26 halocarbons. Of these, some 19 have correlation coefficients versus CCl4 higher than 0.6 (average 0.72). The conclusion that CCl4 is a ubiquitous contaminant of polluted air samples arising from industrial regions of China is obvious. However, the co-locations demonstrated are for emissions of all the substances, not emissions of one against production of the others.

»> Fig. S5 (now Fig. S6 in the revised SI) shows one-to-one correlations between enhanced concentrations of 26 halocarbons, mainly to provide a measure when selecting a reference compound for the interspecies correlation method, which was used to estimate the total emissions of CCl4 in China. It is correct that CCl4 is a ubiquitous contaminant of polluted air samples in China, just like any other long-lived, anthropogenic compounds. Note that HFC-23, which can be expected to be emitted from point sources, showed high correlations of > 0.6 with some 13 species. Therefore, the high one-to-one correlations do not demonstrate co-locations either, because the correlations shown in Fig. S5 (now Fig. S6) were determined from the combined data of all the air masses originated from China. This plot cannot be served to argue that co-locations and thus co-emissions are demonstrated.

Apparently, in China some 90 Gg/y of CCl4 are co-produced with other chloromethanes [Zhang et al., 2015]: this represents only 3% of the total Chinese production of methyl chloride, dichloromethane and chloroform, nevertheless it is the main production source of CCl4. SPARC [2016] and Sherry et al. [2018] were able to account for 13 Gg/y globally as "unreported" emissions from NON-FEEDSTOCK USE. As mentioned above, feedstock emissions were calculated separately and shown to be 2 Gg/y.

»> According to Sherry et al. (2018), the CCl4 production in China from chloromethanes plants and PCE plants were 76.5 Gg yr-1 and 0 Gg yr-1, respectively. The "fugitive emissions" denoted as pathway A by Liang et al. (2016) and Sherry et al. (2018) are calculated to be 2 Gg yr-1, and they included not only "feedstock emissions" but also "process agent emissions".

The evidence presented by the authors is entirely consistent with a higher figure for these unreported emissions from China which would come from industrialised areas, as found. There is, however, no evidence to support the assertion that they arise during production of other chemicals. All that has been proved is that their emissions are geographically co-located with emissions of these other chemicals.

»> As stated earlier, our discussion given in this manuscript was started with a question of how come the banned CCl4 could be co-emitted with CHCl3, CH2Cl2, CH3Cl,

HFC-23, CFC-11 and PCE, and interestingly, resulting source apportionments were consistent with the SPARC report (2016) and Sherry et al. (2018).

The sentence on page 8 lines 31 to 32 "This seems plausible, as evaporative losses of CCl4 during its use as feedstock and/or process agent and from storage reservoirs of factories are easily overlooked and very poorly constrained" is simply wrong. The highest concentration of CCl4 in the air in a storage tank ventilated to the atmosphere is the equilibrium vapour pressure at the temperature of the liquid CCl4 in the tank. In practise the vapour above the liquid is not well mixed, so the vapour pressure represents an absolute maximum. At 20oC, the vapour pressure is such that a full to empty and refill cycle in such a storage tank could release 0.04% of the CCl4. This is well known, easily calculated, and forms the basis of default estimates of fugitive emissions (such as the 0.5% default factor in the IPCC Greenhouse Gas Reporting Guidelines)[IPCC, 2006]. This sentence will become redundant when the discussion of Figure 4 and the conclusions are completely rewritten.

»> Both Sherry et al. (2018) and SPARC mentioned about "potential fugitive leakage during transport and storage" in the section of "Fugitive emissions from usage of CTC" and "some transport leakage, storage leakage" in the section of "CCl4 usage (feedstock, process agent) and destruction", respectively. However, for the sake of argument, we've removed the statements.

In summary, the authors need to accept the difference between emissions and production and to rewrite pages 7 to 9 recognising that their observations relate only to emissions. It might help if they were to co-opt one of the authors of the SPARC report or the Sherry et al. [2018] paper to help them write robust conclusions about the relationships between CCl4 and other emissions. The present text is wrong in almost all respects.

»> See our earlier comment that addressed the discussion in the original pages 7 to 9.

References Carpenter L J and Reimann S (2014) Ozone-depleting substances (ODSs)

and other gases of interest to the Montreal protocol, Ch 1 of Scientific Assessment of Ozone Depletion: 2014, Global Ozone Research and Monitoring Project Report No. 55 (Geneva, Switzerland: World Meteorological Organization). IPCC (Intergovernmental Panel on Climate Change) (2006) IPCC Guidelines for National Greenhouse Gas Inventories Ch 3.10 Fluorochemicals Production, IPCC and IGES, Tsukuba, Japan, 2006 Li, S., Park, M.K., Jo, C.O. Park S. (2017), Emission estimates of methyl chloride from industrial sources in China based on high frequency atmospheric observations, J. Atmos. Chem., 74: 227. https://doi.org/10.1007/s10874-016-9354-4 Simmonds P.G., M. Rigby, A. McCulloch, M.K. Vollmer, S. Henne, J. Mühle, S. O'Doherty, A.J. Manning, P.B. Krummel, P.J. Fraser, D. Young, R.F. Weiss, P.K. Salameh, C.M. Harth, S. Reimann, C.M. Trudinger, L.P. Steele, R.H.J. Wang, D.J. Ivy, R.G. Prinn, B. Mitrevski and D.M. Etheridge (2018), Recent increases in the atmospheric growth rate and emissions of HFC-23 ($CHF_3$) and the link to HCFC-22 ($CHClF_2$) production, Atmos. Chem. Phys., 18, 1–17, 2018,https://doi.org/10.5194/acp-18-1-2018 Sherry D., A. McCulloch, Q. Liang, S. Reimann and P.A. Newman (2018), Current sources of carbon tetrachloride ($CCl_4$) in our atmosphere Environ. Res. Lett. 13024004 SPARC (2016), SPARC Report on the Mystery of Carbon Tetrachloride. Q. Liang, P.A. Newman, S. Reimann (Eds.), SPARC Report No. 7, WCRP-13/2016. Zhang L., Yang W., Zhang L., Li X. (2015), Highly chlorinated unintentionally produced persistent organic pollutants generated during the methanol-based production of chlorinated methanes: A case study in China, Chemosphere, 133, 1-5

Please also note the supplement to this comment: https://www.atmos-chem-phys-discuss.net/acp-2018-220/acp-2018-220-AC4-supplement.pdf

---

## Author Response (AR1)

**Referees' comments on "Toward resolving the budget discrepancy of ozone-depleting CCl4: An analysis of top-down emissions from China"** by Sunyoung Park, Shanlan Li, Jens Mühle, Simon O'Doherty, Ray F. Weiss, Xuekun Fang, Stefan Reimann, Ronald G. Prinn

We thank the referees for their thoughtful and thorough reviews. We are pleased that all the reviewers see our manuscript as a valuable contribution to the field. We have made changes to the manuscript to answer the suggestions of the reviewers and clarified a few points raised in review. We respond to the referee's comments below and a revised version of the manuscript including most of the changes suggested by the reviewers will be submitted to the editor. We thank the reviewers and the editor for their time and effort and appreciate the recommendation for publication in Atmospheric Physics and Chemistry. [In the following, Reviewers' comments are in bold Courier New and our responses and are in Time New Roman font]

**Reviewer comments:**

**Referee #1:**

Paper Summary: This paper uses observations from the Korean Gosan station to identify the location of CCl4 sources and the specific industrial processes involved with the CCl4 emissions. There are two basic techniques used to analyze these data. First, trajectories are used in a source/receptor analysis technique to identify the CCl4 emission locations. The major sources originate in Eastern China. Second, a positive matrix factorization (PMF) analysis technique is used to finger-print the specific sources of  $CCl_4$ . This analysis reveals that the emissions are primarily from chloromethane production  $(CH_3Cl)$ , perchloroethylene production, and fugitive emissions from feedstock usages. These estimates are larger than those from the SPARC CCl4 report, with the fugitive emissions being 10x larger than SPARC!

Review Summary: This is an excellent paper that NEEDS to be published. My overall comments are with regard to improving the writing in the paper and some of the structure.

**Paper Suggestions:**

While the paper mentions the SPARC (2016)  $CCl_4$  report, there ought to be more discussion of how this paper seems to resolve the discrepancy between their emissions based upon measurements. SPARC had a top-down emissions calculation of  $40\pm15$  Gg/y, a hemispheric gradient method of  $30\pm5$  Gg/y, and a regional emission

1

estimates of  $21.4\pm7.5$  Gg/y. The SPARC regional 21.4 Gg/y had a 15 Gg/y contribution from China. The higher estimate herein of 24 Gg/y from China would bring this 21.4 SPARC number up to 30 Gg/y - in precise agreement with the gradient method and within the uncertainties of the 40 Gg/y top down estimate.

>>> Yes, we agree with the reviewer. This is an important point to mention and we have included the following sentences in the second paragraph of the Conclusions section: "Liang et al. (2016) estimated global top-down emissions as  $35 \pm 16$  Gg yr-1, which was an average estimate based on the estimate of  $40 \pm 15$  Gg yr-1 for the new 33-year total lifetime of CCl4 and an independent top-down method using the observed inter-hemispheric gradient in atmospheric concentrations which yielded  $30 \pm 5$  Gg yr-1. The SPARC sum of regional emissions was estimated as  $21 \pm 8$  Gg yr-1, of which Chinese emissions of 15 (10–22) Gg yr-1 contributed 71  $\pm$  33% to the total amount, but this result is still lower than the aggregated top-down values. However, if we employ the higher emission estimate of  $23.6 \pm 7.1$  Gg yr-1, which is largely in agreement with the best global emissions estimate of  $35 \pm 16$  Gg yr-1 determined by Liang et al. (2016)."

The discussion in the summary of the CC14 sources should be broken out with more definitive statements. The SPARC report used industrial estimates to characterize potential sources and this paper provides the [Sherry et al., 2016], \*\*first observational basis\*\* for these sources, but this paper also makes the case that Sherry et al. is perhaps too conservative in their estimates.

>>> This comment is also very helpful. The factors were re-named as (A), (B), (C) and so on, and their descriptions in the section of "Industrial source apportionment of atmospheric CCl4 in East Asia" were also updated to make it easier to compare them to the SPARC report, as suggested by reviewers 1 and 2. The figure legends in Fig. 4 were changed accordingly.

As suggested, we've revised the conclusions to better discuss a link of the industrial sources identified from a factor analysis based on atmospheric observations to the SPARC bottom-up inventory-based estimations.

The revised conclusions now read: "A factor analysis combining the observed concentration enhancements of 18 species was used to identify key industrial sources for CCl4 emissions and to link our atmospheric observation-based top-down identification of potential sources with bottomup inventory-based estimates (e.g., Liang et al., 2016; Sherry et al., 2017). Three major source categories accounting for  $89 \pm 6\%$  of CCl4 enhancements observed at GSN were identified as being related to advertent or inadvertent co-production and escape of CCl4 from CH3Cl production plants (factor (A)),escape during industrial PCE production (factor (C)), and fugitive emissions (factor (B)) from feedstock use for the production of other chlorinated compounds (e.g., CHCl3) and process agent use, and possibly from other chloromethanes use in chemical manufacturing. These sources are largely consistent with the bottom-up CCl4 emissions pathways identified in SPARC (Liang et al., 2016). The SPARC estimate of global CCl4 emissions from chloromethanes and PCE/CCl4 plants (pathway B from Liang et al. (2016) and Sherry et al. (2018)) was 13 Gg yr-1, as the most significant source. Fugitive feedstock/process agent emissions, denoted as pathway A by Liang et al. (2016) and Sherry et al. (2018), were estimated as ~2 Gg yr-1. The emissions contributions from China to pathways B and A were 6.6 Gg yr-1 and 0.7 Gg yr-1, respectively (Liang et al., 2016; Sherry et al., 2018).

If we assume that emission rates from sources correspond to the relative contributions of corresponding source factors to the total Chinese emission rate  $(23.6 \pm 7.1 \text{ Gg yr}^{-1} \text{ for the years } 2011-2015)$ , source factors (A) (CCl4 emissions from chloromethane plants) and (C) (emissions from PCE plants) amount to  $13 \pm 4 \text{ Gg yr}^{-1}$  for China. This is as high as the global bottom-up number of 13 Gg yr-1 for pathway B emissions and more than 50% higher than the Chinese estimate of 6.6 Gg yr-1. This could represent that the ratio of CCl4 emissions from these processes into the atmosphere may be higher than previously assumed, although factor (C) could possibly include the influence of fugitive emissions of CCl4 when using as a chlorination feedstock for PCE production. Furthermore, source factor (B) (fugitive feedstock/process agent emissions) are estimated at ~7 ± 2 Gg yr-1 from China alone, which again contrasts with the Chinese estimate of ~0.7 Gg yr-1 and even with the lower global estimate of only 2 Gg yr-1 for pathway A from Liang et al. (2016) and Sherry et al. (2018)."

The paper is fairly well written, but many of the current paragraphs need to be broken up into more distinct sections or primary thoughts. The extended paragraphs of the current version obscure the thoughts, logic of the paper, and the overall content of the text.

>>> Based on the reviewer's perspective, we realized that discussions should have been better structured in various places in the previous version breaking a long body of paragraph by a single topic. We do think the revised manuscript has been improved according to reviewer's suggestions. Thanks for the reviewer's editorial comments!

For example, the 2nd para of the Introduction (P2, 4-30) talks about top down emissions, bottom up emissions....I would break this up into paras on: 1) top down emissions (4-12); 2) a SPARC bottom up para (12-16); and 3) a discussion of regional emissions.

>>> As the reviewer suggested, we divided this long paragraph into three to make it easier to follow. First, we started with a discussion about the updated bottom-up emissions in the SPARC

report, and introduced the global top-down and hemispheric gradient top-down emissions, pointing out that the revised bottom-up estimate of 25 Gg yr-1 is still lower than the average SPARC-merged top-down emission estimate of  $35\pm16$  Gg yr-1. Then we added the summed regional emissions estimate from Australia, East Asia, U.S. and Western Europe, and mentioned its lowering than the global total and the relative significance of East Asia contribution.

In the 1st para of section 3 (P. 4 line 18 to P.5 line 32 - 46 lines!), there are a broad range of paragraph thoughts. The paragraph starts with a discuss of the interspecies correlation and ends with a thought on an underestimate of Chinese emissions. Please break this up to improve the flow of the text.

>>> We have broken up the original, long paragraph, which is now in section 4 of the revised version, into four paragraphs corresponding to "introduction to an interspecies correlation method", "a reference compound and its emission estimate", "determination of the empirical correlations between the observed enhancements of CCl4 and reference, HCFC-22" and "comparison of the annual CCl4 emissions in China estimated in present study with previous results".

The "Data Overview" section both discusses the data and shows results. I would restructure sections 2 and 3 into: a data, methods, and results sections. The Supporting Information ought to flow better into these data and methods sections.

>>> We have completely restructured section 2 of the manuscript by breaking it up into two subsections (2.1. Measurements of CCl4 at Gosan and 2.2. Results), and one independent section (3. Potential source regions of CCl4 in East Asia). The new section 3 is comprised of three paragraphs: introduction to trajectory statistics as a tool to illustrate the regional distribution of potential CCl4; input data and conditions for calculation; and description of the resulting map of potential source areas. We've also added specific information on corresponding SI text accordingly in the new section 3. Air mass source country classification that had been discussed in the last paragraph of Data overview section in the original manuscript, now moved to the beginning of section 4, as a transitional paragraph to the following country-specific emission analysis.

**Again, break up the single paragraph of the conclusions into short paragraphs. The main messages are lost in this "run-on" paragraph.**

>>> We have re-organized the conclusions with four short paragraphs. We hope this can convey ideas more clearly to readers. For the text revision, please see the earlier response.

Figures are good. For Fig. 4, put some vertical lines on the plot to see how the bars line up with chemical names at the bottom.

>>> Done

Fig. S5. What are the colors for? Do they indicate statistical significance?

>>> We now say in the figure caption: "The colors by shade indicate statistical significance."

**Referee #2:**

Park et al presented a top-down emissions estimate of CC14 from East Asia based on high frequency surface measurements of halocarbons at the Gosan sites. This paper is timely. Results presented in this paper provide crucial pieces of information that closes the CCl4 global budget as well as providing the observational evidence that unreported CC14 atmospheric emissions during chloromethans and PCE production. However, the writing in many places can use some improvements. I recommend the authors go through the entire manuscript thoroughly to improve the clarity and accuracy. The paper should be published in ACP after the following comments are addressed.

**1. P1 L15, "the 2010" -> "a 2010"**

**>>> Done**

2. P2 L5-7. You should state that the global top-down emissions are derived based on both the CCl4 lifetimes and the observed global decline rate.

>>> A point well-taken. We have changed it to what the reviewer suggested (underlined words are the edits): "To verify these bottom-up estimates, independent top-down CCl4 emission studies have used the total lifetime of CCl4 with atmospheric observations (i.e., the observed decline rate of CCl4 concentrations) and atmospheric transport models to derive "top-down" emission estimates."

**3. P2 L9. The global emissions number from Liang et al, 2014 was 39Gg/yr, not 30Gg/yr.**

>>> We realized from the reviewer's comments that the citation was incorrect. The 39 Gg/yr emission from Liang et al. (2014) had been updated into the value of 30 Gg/yr with the new 33-year lifetime of CCl4 in the SPARC report (Liang et al., 2016).

So we've changed the original sentence into the following to clarify the updated estimate: "A recent top-down study based upon the observed temporal trend and inter-hemispheric gradient of atmospheric CCl4 (Liang et al., 2014) consistently derived global CCl4 emissions of  $30 \pm 5$  Gg yr-1 from 2000 to 2012 when using the newly determined relative strength of oceanic sink versus soil loss (Liang et al., 2016)."

**4. P2 L11-12. I am not sure why you say "unidentified sources**

**and/or unreported anthropogenic emissions". CCl4 is a predominantly man-made compound, therefore the emissions sources are anthropogenic.**

>>> Agreed. For clarification the word "anthropogenic" has been edited into "industrial". We think unidentified old, contaminated soils and/or facilities can be "unidentified sources" here.

5. In many places, need to change the "," after the references to ";".

>>> Corrected

6. P2 L27-30. You need to merge these two sentences and present the results from these studies in a less confusing way with a correct referencing style. In the present form, it is hard for the readers to figure out from which studies the 4.3 and 5.2 Gg/yr were from.

>>> The sentences have been merged and edited to clarify that those numbers were updated as new bottom-up emission estimates (underlined words are the edits): "Most recently, Bie et al. (2017) published post-2010 bottom-up emission estimates for China of 4.3 (1.9–8.0) Gg yr-1 in 2011 and 5.2 (2.4–8.8) Gg yr-1 in 2014, which updated the previous zero emissions estimate (Wan et al., 2009) by including the conversion of C2Cl4 emissions to CCl4 as well as the source of CCl4 from coal combustion smog.

7. P2 L30. Change to "8-year continuous high frequency, high precision atmospheric CCl4 concentrations measured..."

>>> Changed

8. P3 L2. Change "below the " to "to the south of ..."

>>> Changed

9. P3 L7. I am not sure what do you mean by "well situated to allow monitoring of long-range transport from the surrounding region". Is this because of elevation or it is in remote clean ocean? By surrounding region, what regions are you referring to? China? The Korean Peninsula? Please clarify. >>> We now provide more explicit description of the station in the Supplementary Information as well as give more information in the figure caption (Fig. S1): "Gosan station (GSN, 33.25°N, 126.19°E, Jeju Island, Korea) is located on the boundary between the Pacific Ocean and the Asian continent (Fig. S1), which experiences a warm wet East Asian Summer Monsoon, a cold dry winter, and distinct seasonal wind patterns (strong northern winds in winter and a southern influence during summer). These wind patterns are favorable for monitoring air masses passing through East Asia, particularly through China and Korea. Clean background conditions are observed when a clean stream of air flows in directly from northern Siberia in winter and during transport of southerly oceanic winds in summer (Fig. S2)."; "**Fig. S1**. Gosan AGAGE (Advanced Global Atmospheric Gases Experiment) station is located on a 72-m cliff (air intake elevation: 89 m above sea level) on the remote south-western tip of Jeju Island, 100 km south of the Korean peninsula, allowing for monitoring of long-range air mass transport from the surrounding region."

**10. P3 L10. Please include the actual values than just say "high-precision and high frequency"**

>>> The data frequency has been given as "every two hours from 2008 to 2015" and the experimental precision has also been stated in the sentence: "Precisions (1 $\sigma$ ) derived from repeated analysis (n = 12) of a working standard of ambient air are better than 1 % of background atmospheric concentrations for all the compounds, e.g.  $\pm 0.8$  ppt (1 $\sigma$ ) for 85.2 ppt of CCl4." "

**11. P3 L22. You need to define what do you mean by "baseline values". This is jargon.**

>>> We have added the following text in parentheses: "(i.e., background values representing regional clean conditions without regional/local pollution events, black dots)".

**12. P4 L6-7. It would be good to add references here.**

>>> We've added the website http://eng.chinaiol.com/, where the locations of the main factories producing HFCs, HCFC-22 and fluorocarbons are given. The locations were also denoted in Fig. S9.

**13. P7 L11. It is interesting that CFC-11 showed up in the source factor. Does this indicate that CFC-11 is also produced in the CM plants?**

>>> Given the fact that CFC-11 can be readily produced by the reaction of by-produced impurity, CCl4 with HF, the observed high contribution of CFC-11 in the fugitive emissions group is

explainable in association with production of chloromethanes and their feedstock use for fluorinated compounds. For further comments regarding recent enhancements of CFC-11 observed at Gosan, please see the last response below.

14. P7. It will be of great value to CCl4 source identification to link the discussions in the source factors to the industrial production, usage, and potential emissions pathway in Sherry et al. (2017). Such a discussion will help to build link from bottom-up inventory-based estimate to atmospheric observation based top-down estimate.

>>> Agreed. According to the reviewer's suggestion, we've revised the conclusions to better discuss a link of the industrial sources identified from a factor analysis based on atmospheric observations to the SPARC bottom-up inventory-based estimations.

The revised conclusions now read: "A factor analysis combining the observed concentration enhancements of 18 species was used to identify key industrial sources for CCl4 emissions and to link our atmospheric observation-based top-down identification of potential sources with bottomup inventory-based estimates (e.g., Liang et al., 2016; Sherry et al., 2017). Three major source categories accounting for  $89 \pm 6\%$  of CCl4 enhancements observed at GSN were identified as being related to advertent or inadvertent co-production and escape of CCl4 from CH3Cl production plants (factor (A)), escape during industrial PCE production (factor (C)), and fugitive emissions (factor (B)) from feedstock use for the production of other chlorinated compounds (e.g., CHCl3) and process agent use, and possibly from other chloromethanes use in chemical manufacturing. These sources are largely consistent with the bottom-up CCl4 emissions pathways identified in SPARC (Liang et al., 2016). The SPARC estimate of global CCl4 emissions from chloromethanes and PCE/CCl4 plants (pathway B from Liang et al. (2016) and Sherry et al. (2018)) was 13 Gg yr-1, as the most significant source. Fugitive feedstock/process agent emissions, denoted as pathway A by Liang et al. (2016) and Sherry et al. (2018), were estimated as ~2 Gg yr-1. The emissions contributions from China to pathways B and A were 6.6 Gg yr-1 and 0.7 Gg yr-1, respectively (Liang et al., 2016; Sherry et al., 2018).

If we assume that emission rates from sources correspond to the relative contributions of corresponding source factors to the total Chinese emission rate  $(23.6 \pm 7.1 \text{ Gg yr}^{-1} \text{ for the years } 2011-2015)$ , source factors (A) (CCl4 emissions from chloromethane plants) and (C) (emissions from PCE plants) amount to  $13 \pm 4 \text{ Gg yr}^{-1}$  for China. This is as high as the global bottom-up number of 13 Gg yr-1 for pathway B emissions and more than 50% higher than the Chinese estimate of 6.6 Gg yr-1. This could represent that the ratio of CCl4 emissions from these processes into the atmosphere may be higher than previously assumed, although factor (C) could possibly include the influence of fugitive emissions of CCl4 when using as a chlorination feedstock for PCE production. Furthermore, source factor (B) (fugitive feedstock/process agent emissions) are estimated at ~7 ± 2 Gg yr-1 from China alone, which again contrasts with the Chinese estimate of ~0.7 Gg yr-1 and even with the lower global estimate of only 2 Gg yr-1 for

pathway A from Liang et al. (2016) and Sherry et al. (2018)."

15. Figure 3 and related discussions.

(1) I wonder if part of the difference between the Vollmer et al., 2009 and this study is due to the location of Gosan vs. Shandianzi. The location of Gosan captures most of the outflow from the industrial central and south China, where all the CC14 production industries are located (as suggested by Figure 2), while Shandianzi captures mostly the air influenced by N. China, without much CM production. Should consider add a related discussion on this in the manuscript.

>>> Yes, this is an important point to mention. We agree with the reviewer that difference in the location of monitoring sites and thus in their footprint distributions of compounds of interest must be one of potential reasons for discrepancies found between emission estimates derived from different monitoring sites.

Interestingly, however, the CCl4 emission rate of  $16.8 \pm 5.6$  Gg yr-1 in 2008 we derived in this study was statistically consistent with the 2007 emission rate of 15 (10–22) Gg yr-1 given in Vollmer et al. (2009) within their uncertainties. The agreement could be coincidental, but it could also be consistent with the fact that even though the CM-related production facilities are more likely located in industrial central and south China (Fig 2 and Fig S9), the increase in both the feedstock production sector of CCl4 and emissions from CCl4 by-production was reported only since 2011, i.e. post-2010 (Bie et al., 2017: see Fig. 2 in the paper).

In this respect, it is possible that the 2007 emission estimate derived from Shandianzi and the 2008 estimate from and Gosan were not much different, even if Shandianzi is known to capture mostly the air masses influenced by north China – covering most down to Shandong and Anhui for CCl4 (Vollmer et al., 2009) and to Jiangsu and Anhui for CO (An et al., 2014), and thus could possibly miss the influences from Henan, Hubei, and Guangdong provinces.

Therefore, it seems that further discussion about potential differences in emissions estimate for CCl4 between Gosan vs. Shandianzi, particularly in relation to the location of CCl4 emission sources can be made when further analysis on the CCl4 data and results of post-2010 from Shandianzi are published.

**Reference:**

An, X., Yao, B., Li, Y., Li, N., Lingxi Zhou, L.: Tracking source area of Shangdianzi station using Lagrangian particle dispersion model of FLEXPART, Meteorol. Appl. 21: 466–473, 2014.

(2) The covariance of CFC-11 and CMs (source factor 2) is very interesting. Does this mean CFC-11 is also an intendended by-product during the industrial process and the recent increase in CFC-11 unreported emissions (Montzka et al. 2018) is to some

**extent linked to the CCl4 emissions increase in China between 2012-2016?**

>>> As we noted in response to the comment above regarding the high contribution of CFC-11 shown in the fugitive emissions group, CFC-11 can be readily produced by the reaction of byproduced impurity, CCl4 with HF and thus it would be possible that the observed high contribution of CFC-11 in the fugitive emissions group could be association with production of chloromethanes and their feedstock use for fluorinated compounds, whether it is intended or not Recent increase in unreported CFC-11 emissions discussed in Montzka et al. (2018) is indeed consistent with recent enhancements in CFC-11 pollution signals observed at Gosan (see figures below). It would also be possible that these enhancements might be associated with production of many fluorinated compounds using chloromethanes as feedstock and thus with persistent CCl4 emissions in East Asia, as shown in this study.

This reviewer's question is one of the most important issues these days. So, if allowed we'd like to complete a separate analysis for CFC-11enhancements at Gosan and address this issue further in another manuscript.

**Referee #3:**

There has been a long-standing mystery of why the atmospheric concentration of carbon tetrachloride has declined much slower than predicted after its use was banned by the Montreal Protocol. The SPARC (2016) report resolved only part of this mystery by assessing a slightly longer atmospheric lifetime and by increasing estimates of industrial bottom-up emissions. However, a reconciliation of the top-down and bottom-up estimates was not achievable unless the error bars were stretched to their limits.

The present study by Park et al. utilizes high precision measurements of a suite of halocarbons at a background air monitoring station at Gosan, South Korea, to identify the origins of large fugitive emissions of CCl4 and to estimate their overall emission rates between 2008-2015. The analysis determines that emissions from heavily industrialized regions of China can account for roughly 24 + 7 Gg/yr CCl4 between 2011 and 2015 instead of the 4-5 Gg/yr reported bottom-up emissions rates. Surprisingly, emission rates do not seem to have declined over this time period. The additional 19 Gg/yr of fugitive emissions from China would account for over half of the global CC14 emissions, and perhaps be enough to resolve the remaining mystery of carbon tetrachloride. Thus, this paper represents a very important scientific advance indeed.

The atmospheric measurements are of high quality and the method back trajectories combined with of using air empirical correlations with a reference compound (HCFC-22) is supported by an independent derivation of HCFC-22 emissions that agrees with prior estimates. The industrial source apportionment using the Positive Matrix Factorization (PMF) model yielded several strong relationships, pointing to multiple sources of CC14 associated largely with emissions with other compounds. The interpretation is that the fugitive emissions are occurring at the factory level during production of various chlorocarbons. This seems highly plausible, as the production of these compounds are colocated, whereas the consumption of these compounds are expected to be more widely distributed.

Overall, the writing and figures are clear, and the methodology maximizes the functionality of a high quality dataset. I

12

encourage the publication of this important work, with only a few minor edits suggested below.

1. Pg 2, line 6. The relevant soil sink reference is: Rhew & Happell, 2016, not Rhew et al., 2008.

>>> Changed. Thanks much!

2. Pg 3, line 10. Here it would be helpful to have a reference or more description about the Gosan station. A brief description of the sample intake line, its height and its proximity to other major landscape features would be helpful details.

>>> We now provide more explicit description of the station in the Supplementary Information as well as give more information in the figure caption (Fig. S1): "Gosan station (GSN, 33.25°N, 126.19°E, Jeju Island, Korea) is located on the boundary between the Pacific Ocean and the Asian continent (Fig. S1), which experiences a warm wet East Asian Summer Monsoon, a cold dry winter, and distinct seasonal wind patterns (strong northern winds in winter and a southern influence during summer). These wind patterns are favorable for monitoring air masses passing through East Asia, particularly through China and Korea. Clean background conditions are observed when a clean stream of air flows in directly from northern Siberia in winter and during transport of southerly oceanic winds in summer (Fig. S2)."; "**Fig. S1**. Gosan AGAGE (Advanced Global Atmospheric Gases Experiment) station is located on a 72-m cliff (air intake elevation: 89 m above sea level) on the remote south-western tip of Jeju Island, 100 km south of the Korean peninsula, allowing for monitoring of long-range air mass transport from the surrounding region."

**3. Pg 3, line 20. The authors should specify that the remote background station in the Northern Hemisphere is Mace Head, Ireland.**

>>> We've specified Mace Head station as a NH remote monitoring site (underlined words are the edits): "The background concentrations at GSN were determined using the statistical method detailed in O'Doherty et al. (2001), and they agree well with those observed at the Mace Head station (53°N, 10°W) in Ireland (which is representative of a remote background monitoring station in the Northern Hemisphere) and are declining at a similar rate to the global trend (Fig. S4)."

On a related note, it appears that no other AGAGE station comes anywhere close to the pollution level events that Gosan station experiences. Expressing this, perhaps in a quantitative way (standard deviation?) would add to the argument that the Gosan station is uniquely situated among the network to capture the primary region of fugitive emissions. After seeing the data published online from all the other stations, it seems clear that this is so.

>>> A point well-taken. We've revised the description about the time series plot of the atmospheric CCl4 concentrations observed at Gosan (Fig. 1) in the section 2 (underlined words are the edits): "The 8-year observational record of CCl4 analyzed in this study is shown in Fig. 1. It is apparent that pollution events (red dots) with significant enhancements above "background" levels (black dots) occurred frequently, resulting in daily variations of observed concentrations with relative standard deviations (RSDs) of 4–20% (in contrast to the RSDs of 0.1–1.5% shown in all the remote stations operated under the AGAGE program). These results clearly imply that CCl4 emissions are emanating from East Asia."

4. Section 4. Although the time periods may differ, it may be results useful compare these with some ground based to measurements within China that are closer to the source regions. For example, prior studies have found very high concentrations of halocarbons in the Pearl River Delta region of China. Zhang et al., (JGR 115, D15309,2010) measured elevated concentrations in 2007 and report "The high correlation between CCl4 and CFCs suggests that this source was more related to the production than the consumption of refrigerants." How important is the Delta region compared Pearl River to other regions in the present study? It is difficult to assess based on the maps.

>>> This is a good suggestion. The Pearl River Delta (PRD) region denoted by blue circles in Fig. S9 shown below is one of important source regions in China.